# Optimizing Anytime Reasoning
# via Budget Relative Policy Optimization

**Penghui Qi[12], Zichen Liu[12], Tianyu Pang[1], Chao Du[1], Wee Sun Lee[2], Min Lin[1]**
[1]Sea AI Lab    [2]National University of Singapore
 https://github.com/sail-sg/AnytimeReasoner

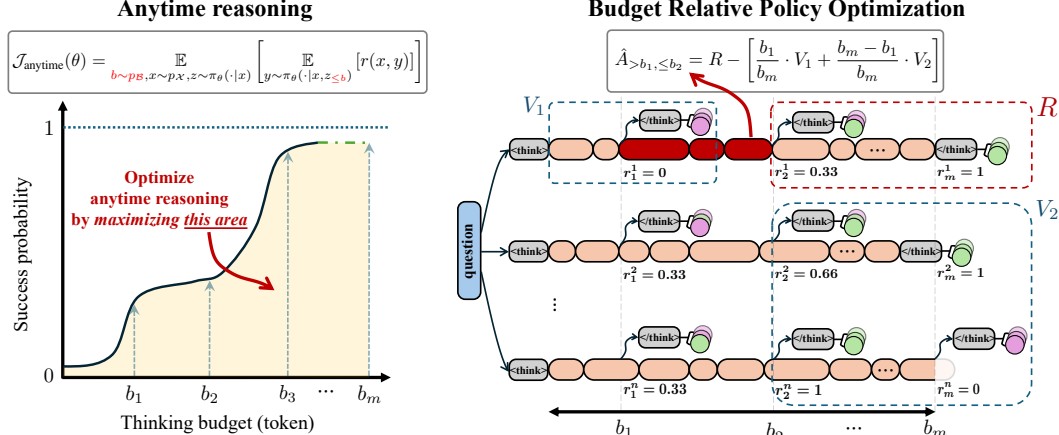

Figure 1: **Left**: We optimize *anytime reasoning* by sampling thinking budgets from a prior distribution $p_{\mathcal{B}}$ and maximizing the rewards at sampled budgets to push up the area under the curve. This objective naturally introduces *verifiable dense rewards* into the thinking process. **Right**: Budget Relative Policy Optimization (BRPO) leverages these dense rewards to improve advantage estimation via the Monte Carlo return ($R$) and an interpolated baseline that combines current progress ($V_1$) and the average return within the rollout group ($V_2$).

## Abstract

Scaling test-time compute is crucial for enhancing the reasoning capabilities of large language models (LLMs). Existing approaches typically employ reinforcement learning (RL) to maximize a verifiable reward obtained at the end of reasoning traces. However, such methods optimize only the final performance under a large and fixed token budget, which hinders efficiency and flexibility in both training and deployment. In this work, we present **AnytimeReasoner**, a novel framework for optimizing reasoning performance under varying thinking budget constraints. To achieve this, we truncate the complete thinking process to fit within sampled token budgets from a prior distribution, compelling the model to summarize the optimal answer for each truncated thinking for verification. This introduces **verifiable dense rewards** into the reasoning process, facilitating more effective credit assignment in RL optimization. We then optimize the thinking and summary policies in a decoupled manner to maximize the cumulative reward. Additionally, we introduce a novel variance reduction technique, **B**udget **R**elative **P**olicy **O**ptimization (**BRPO**), to enhance the robustness and efficiency of the learning process when reinforcing the thinking policy. Empirical results in mathematical reasoning tasks demonstrate that our method consistently outperforms GRPO across all thinking budgets under various prior distributions, enhancing both training and token efficiency.

39th Conference on Neural Information Processing Systems (NeurIPS 2025).

# 1 Introduction

OpenAI o1 [OpenAI, 2024] and DeepSeek-R1 [Guo et al., 2025] have shown that scaling test-time compute via RL is crucial for LLM reasoning. This involves an extensive thinking process using the chain of thought (CoT) [Wei et al., 2022] before producing an answer. RL is then employed to maximize the outcome reward provided by a rule-based verifier to check the correctness of the generated answer. While RL for LLM reasoning is an active area of research, most existing work focuses on optimizing final performance based on the complete thinking process. This approach can be inefficient in both training and deployment, as long CoTs are costly, especially for online services.

In our work, we focus on **optimizing anytime reasoning for LLMs via RL**. This is conceptually similar to the *anytime algorithms* introduced in Dean and Boddy [1988], Zilberstein and Russell [1995], where the system can be interrupted at any point during computation, providing the best possible solution so far and is expected to improve the solution quality when more resources are allocated. Concretely in LLM reasoning, we assume the thinking process can be interrupted at any time, and the model should be able to summarize the best solution from incomplete thinking. This capability can significantly extend the serving capacity for online services with limited computing resources. When there are too many requests to handle, the service can choose to interrupt in-progress requests once the thinking length is able to give sufficient accuracy, reserving longer thinking with better accuracy when resources are available. Moreover, users may want to control the thinking budget as in Gemini 2.5[Comanici et al., 2025], but the optimal budget is often agnostic. Compared to budget-aware reasoning[Han et al., 2024], our design supports an economical strategy by incrementally increasing the budget, as it allows for continued thinking and reuses the computation already spent.

To achieve optimal performance for anytime reasoning, we propose **sampling the thinking budget from a prior distribution** while learning, rather than using a fixed, large budget as in prior work [Liu et al., 2025, Zeng et al., 2025, Luo et al., 2025]. This approach makes the model performance robust to potential interruptions in the thinking process, while incentivizing it to reach correct answers more efficiently. By achieving a balance between token efficiency and thorough exploration [Qu et al., 2025], these models are also able to obtain better performance when given larger budgets.

We investigate how to efficiently train LLMs with RL under sampled thinking budgets. By forcing the model to summarize the answers at predefined thinking budgets (drawn from the support of the prior distribution), we introduce **verifiable dense rewards** into the reasoning process. These rewards provide richer signals and better credit assignment during training [Qu et al., 2025, Cui et al., 2025a]. We also propose **a novel variance reduction technique termed Budget Relative Policy Optimization (BRPO) that advances beyond GRPO** [Shao et al., 2024] to improve training stability and efficiency under this dense reward framework. As illustrate in Figure 1 (right), we leverage rewards at previous budgets to compute the advantage function, combining with the average return of a group of reasoning trajectories. Empirically, we observe that generating a high-quality summary is critical for both final and anytime performance. Thus, we **decouple the optimization of the thinking and summary policies**, always sampling from a uniform distribution to derive a better summary policy, thereby improving training efficiency.

We term our overall framework as *AnytimeReasoner*. Experimental results demonstrate that **AnytimeReasoner consistently surpasses GRPO in both final and anytime performance**. We conduct extensive ablation studies to evaluate the impact of each component. By independently incorporating decoupled optimization, variance reduction, and budget sampling into GRPO, we observe significant performance enhancements, underscoring the effectiveness of our methods. Notably, even when merely using the maximum token budget (without budget sampling), our method still outperforms GRPO in both standard and anytime reasoning, highlighting the robustness of our approach.

# 2 Methodology

In a training paradigm similar to R1-Zero [Guo et al., 2025], the model is tasked with generating a comprehensive CoT within a designated "thinking box" upon receiving a question. Subsequently, the model summarizes the answer based on this thinking process. A rule-based reward is then calculated according to the summarized answer. The RL objective is to maximize the expected reward:

$$\mathcal{J}(\theta) = \mathbb{E}_{\underbrace{x \sim p_\mathcal{X}}_{\text{question}}} \mathbb{E}_{\underbrace{z \sim \pi_\theta(\cdot|x)}_{\text{thinking process}}} \mathbb{E}_{\underbrace{y \sim \pi_\theta(\cdot|x,z)}_{\text{answer}}} [r(x,y)] \tag{1}$$

where $x$ represents the question, $z$ denotes the thinking process, $y$ is the summarized answer, and $r(x, y)$ is the reward function.

In previous studies [Zeng et al., 2025, Liu et al., 2025, Luo et al., 2025], the generation of thinking process and summary are typically sampled together. If the thinking process exceeds the predefined generation limit, the response is considered a negative sample. We contend that this approach is impractical, particularly in online services where a valid summary should be provided even if the thinking process is incomplete. We propose decoupling the generation of the thinking process and its summary, allocating separate token budgets for each. When the thinking process is halted due to budget constraints, we insert ellipses followed by a *</think>* to prompt the model to produce a summary (see Appendix A), similar to Muennighoff et al. [2025] and Qu et al. [2025].

To differentiate between the thinking and summary policies, we denote the thinking policy as $\pi_\theta$ and the summary policy as $\pi_\phi$. By defining $r_\phi(x, z) = \mathbb{E}_{y \sim \pi_\phi(\cdot | x, z)}[r(x, y)]$, the objective can be expressed as:

$$\mathcal{J}(\theta, \phi) = \mathbb{E}_{x \sim p_\mathcal{X}, z \sim \pi_\theta(\cdot | x)}[r_\phi(x, z)]. \qquad (2)$$

Given that $|y| \ll |z|$, multiple summaries can be sampled to better estimate the expected reward for each thinking process, while incurring only a small computational overhead.

## 2.1 Optimizing Anytime Reasoning

Test-time scaling [OpenAI, 2024] is crucial for enhancing the reasoning capabilities of LLMs. This concept operates on the premise that increased computational effort during the reasoning process generally leads to better performance. However, in typical RL training setups like R1-Zero-like [Guo et al., 2025], the performance on anytime reasoning is not guaranteed. The reward evaluation is based on the entire thinking process, lacking insight into whether incremental thinking consistently improves performance [Qu et al., 2025].

To optimize anytime reasoning, we propose sampling the thinking budget from a prior distribution rather than using a fixed token budget. Let $b$ represent the token budget for thinking, sampled from a prior distribution $p_\mathcal{B}$ over a set of increasing budgets $\{b_1, \ldots, b_m\}$ ($P_j = p_\mathcal{B}(b = b_j)$ for simplicity). The anytime reasoning objective is:

$$\mathcal{J}_{\text{anytime}}(\theta, \phi) = \mathbb{E}_{b \sim p_\mathcal{B}, x \sim p_\mathcal{X}, z \sim \pi_\theta(\cdot | x)}[r_\phi(x, z_{\leq b})] = \mathbb{E}_{x \sim p_\mathcal{X}, z \sim \pi_\theta(\cdot | x)}\left[\sum_{j=1}^{m} P_j r_\phi(x, z_{\leq b_j})\right], \qquad (3)$$

where $z_{\leq b}$ is the truncated thinking process at length of the token budget $b$,

$$z_{\leq b} = \begin{cases} z, & \text{if } b \geq |z| \\ \text{truncate}(z, b), & \text{if } b < |z| \end{cases}.$$

Instead of focusing solely on the final score based on the entire thinking process as in standard reasoning task, we maximize the expected score over all possible budgets with distribution $p_\mathcal{B}$. As illustrated in Figure 1, this is akin to maximizing the area under the score curve when $p_\mathcal{B}$ is a uniform distribution across every token budget. However, evaluating for all token budgets is impractical and unnecessary, so we evaluate the score only at a small predefined budget support (with $m \leq 8$ in our experiments).

It is important to note that this approach transforms the problem into a dense reward framework, introducing verifiable dense rewards for each thinking budget. This facilitates better credit assignment during RL training and enhances the identification of each component's contribution to a successful reasoning process. As illustrated in Figure 2, the dense rewards for budgets prior to reaching a correct answer are low. However, the cumulative return is relatively higher if the reasoning process ultimately arrives at a correct answer. In contrast, the cumulative return after the first correct answer is relatively low, localizing and highlighting the tokens that contributed to the initial correct answer. This approach is distinct from typical sparse reward RL training for standard reasoning tasks, where all tokens receive the same return. Such sparse reward structures typically lead to unstable and inefficient RL training, while our dense reward approach provides more informative learning signals throughout the entire reasoning process.

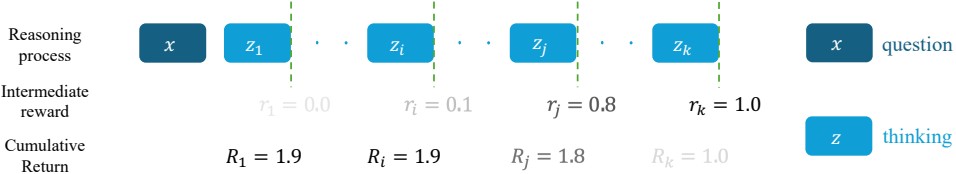

Figure 2: By introducing dense rewards, we achieve better credit assignment during RL training. We assume a uniform distribution over thinking budgets and omit the probability for simplicity.

**Relation to Standard Reasoning Tasks**  A larger thinking budget is supposed to yield better performance in expectation. Since $z_{\leq b}$ is always a prefix of $z$, the optimal summary policy $\pi_{\phi^*}$ should satisfy:

$$\mathbb{E}_{z \sim \pi_\theta(\cdot|x)} \left[ r_{\phi^*}(x, z_{\leq b}) \right] \leq \mathbb{E}_{z \sim \pi_\theta(\cdot|x)} \left[ r_{\phi^*}(x, z) \right], \tag{4}$$

for any $b$ and $x$. Then we have:

$$\mathcal{J}_{\text{anytime}}(\theta, \phi^*) \leq \mathcal{J}(\theta, \phi^*) \tag{5}$$

This justifies the anytime reasoning objective as a lower bound of the standard reasoning objective. Therefore, maximizing performance in anytime reasoning should also enhance performance in standard reasoning tasks. In an extreme case where $P_m = 1$ (training only with full reasoning length), $\mathcal{J}_{\text{anytime}}$ falls back to the standard reasoning objective $\mathcal{J}$. For detailed proof, refer to Appendix C.

## 2.2   Budget Relative Policy Optimization

By defining $j_t = \arg\min_j b_j \geq t$, which represents the nearest token budget after $t$, the gradient for the thinking policy can be computed as follows:

$$\nabla_\theta \mathcal{J}_{\text{anytime}}(\theta, \phi) = \mathbb{E}_{x \sim p_\mathcal{X}, z \sim \pi_\theta(\cdot|x)} \left[ \sum_{t=1}^{|z|} \nabla_\theta \log \pi_\theta(z_t|x, z_{<t}) \left( R(x, z, j_t) - V(x, z_{<t}) \right) \right], \tag{6}$$

where

$$R(x, z, j_t) = \sum_{j=j_t}^{m} P_j r_\phi(x, z_{\leq b_j}),$$

and $V(x, z_{<t})$ is the variance reduction term, which should be a function correlated to $R(x, z, j_t)$ but invariant with respect to $z_t$.

Typically, we set $V(x, z_{<t}) = \mathbb{E}_{z_{\geq t} \sim \pi_\theta(\cdot|x, z_{<t})} [R(x, [z_{<t}, z_{\geq t}], j_t)]$, representing the expected future return [Sutton and Barto, 2018]. In traditional RL, GAE[Schulman et al., 2015] is often used by estimating this value with a critic model. However, training a critic model for LLM can be both costly and noisy [Guo et al., 2025]. An alternative is sampling-based approach, as in VinePPO [Kazemnejad et al., 2024] and Remax [Li et al., 2023], but this requires significant additional computation across all thinking budgets. Group-based methods, such as GRPO [Shao et al., 2024] and RLOO [Ahmadian et al., 2024], treat generation as a bandit and use the average score of multiple responses for variance reduction. However, they are unsuitable in our scenario due to the presence of dense rewards.

In LLM generation, newly sampled tokens (actions) are consistently appended to the existing context (states). This implies that the current context ($z_{<t}$) always serves as a prefix for any future context ($[z_{<t}, z_{\geq t}]$). This unique property distinguishes it from traditional RL but is often overlooked. Assuming a perfect summary policy that consistently extracts the best answer from the thinking process, the reward should increase monotonically with the number of generated tokens, satisfying $r_\phi(x, z_{<t}) \leq r_\phi(x, [z_{<t}, z_{\geq t}])$. Consequently, the current reward $r_\phi(x, z_{<t})$ is correlated with any future reward $r_\phi(x, [z_{<t}, z_{\geq t}])$, particularly when $t$ is large enough to yield a correct answer or when $|z_{<t}| \gg |z_{\geq t}|$. This correlation justifies its use as a suitable baseline for variance reduction.

Building on this insight, we introduce **B**udget **R**elative **P**olicy **O**ptimization (**BRPO**) for efficient variance reduction. Specifically, we employ the following variance reduction term:

$$V_1 = \frac{\sum_{j=1}^{j_t-1} \lambda^{j_t-j} r_\phi(x, z_{\leq b_j})}{\sum_{j=1}^{j_t-1} \lambda^{j_t-j}} \sum_{j=j_t}^{m} P_j, \tag{7}$$

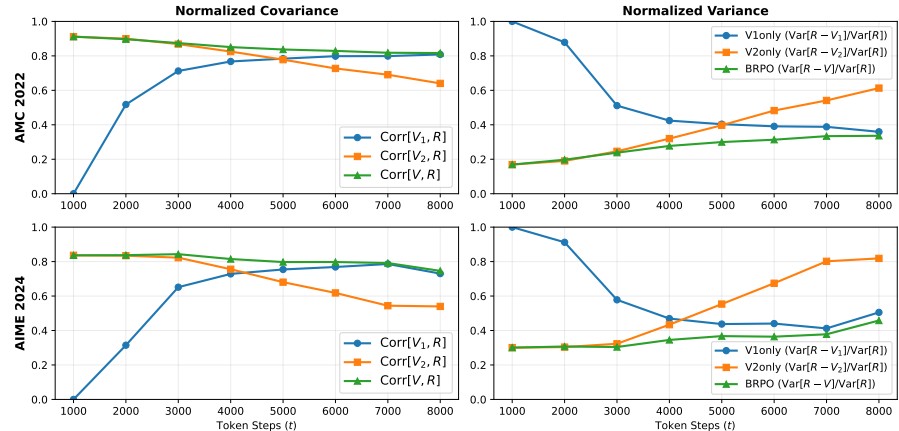

Figure 3: **Left**: The correlation coefficient of $V_1$ and $V_2$ with $R(x, z, j_t)$. **Right**: The normalized variance of our BRPO. We evaluate the R1-Distill-1.5B model under the scenario where $\lambda = 0.5$, and $p_{\mathcal{B}}$ is a uniform distribution over $\{1000, 2000, ..., 8000\}$.

where the evaluated scores at previous budgets, weighted by a discount factor $\lambda$, serve as the reward baseline (highlighted in red), and are multiplied by the sum of probabilities after $j_t$ to align with the scale of $R(x, z, j_t)$.

As illustrated in Figure 3, when $t$ is small, the effectiveness of $V_1$ may diminish because a short thinking process $z_{<t}$ provides limited information. In such cases, we apply a variant of GRPO as a complement. We sample a set of thinking processes $\{z^1, z^2, \ldots, z^G\}$ and compute:

$$V_2 = \frac{1}{G} \sum_{i=1}^{G} R(x, z^i, j_t), \tag{8}$$

which represents the expected return after $j_t$ given the question $x$. Note that the correlation between $V_2$ and $R(x, z, j_t)$ decreases as $t$ increases, as shown in Figure 3, due to differing prefixes ($z_{<t}$) in these thinking processes.

By combining $V_1$ and $V_2$, the overall variance reduction term is:

$$V(x, z_{<t}) = \frac{j_t - 1}{m} V_1 + \frac{m - j_t + 1}{m} V_2. \tag{9}$$

As demonstrated in Figure 3, our BRPO significantly outperforms GRPO in reducing variance, especially when the thinking is long.

## 2.3 Decoupled Optimization for Thinking and Summary

In a rigorous derivation, the optimization of thinking and summary policies should share the same prior budget distribution $p_{\mathcal{B}}$. However, an optimal summary policy is crucial when the thinking process is incomplete, and its effectiveness is significantly influenced by $p_{\mathcal{B}}$. An imbalanced prior distribution can lead to suboptimal summary policy. To achieve a robust anytime reasoning performance, we decouple the optimization of thinking and summary policies by using a different budget distribution, $p'_{\mathcal{B}}$, for the summary policy. The decoupled gradient of the summary policy with respect to the anytime reasoning objective 3 can be computed as follows:

$$\nabla_\phi \mathcal{J}_{\text{anytime}}(\theta, \phi) = \mathbb{E}_{x \sim p_{\mathcal{X}}, z \sim \pi_\theta(\cdot|x)} \left[ \sum_{j=1}^{m} P'_j \mathbb{E}_{y \sim \pi_\phi(\cdot|x, z_{\le b_j})} \left[ \nabla_\phi \log(\pi_\phi(y|x, z_{\le b_j})) r(x, y) \right] \right]. \tag{10}$$

In our experiments, we set $p'_{\mathcal{B}}$ as a uniform distribution over the budget support $\{b_1, \ldots, b_m\}$. We employ a distinct approach to optimize the summary policy. Specifically, for each question $x$ and thinking process $z_{\le b_j}$, we sample a group of summaries and use GRPO to stabilize the optimization.

Typically, a shared model ($\phi = \theta$) is used for both thinking and summary policies. In such cases, the overall gradient is:

$$\nabla_\theta \mathcal{J}_{\text{anytime}}(\theta) = \nabla_\theta \mathcal{J}_{\text{anytime}}(\theta, \phi) \big|_{\phi=\theta} + \nabla_\phi \mathcal{J}_{\text{anytime}}(\theta, \phi) \big|_{\phi=\theta}.$$

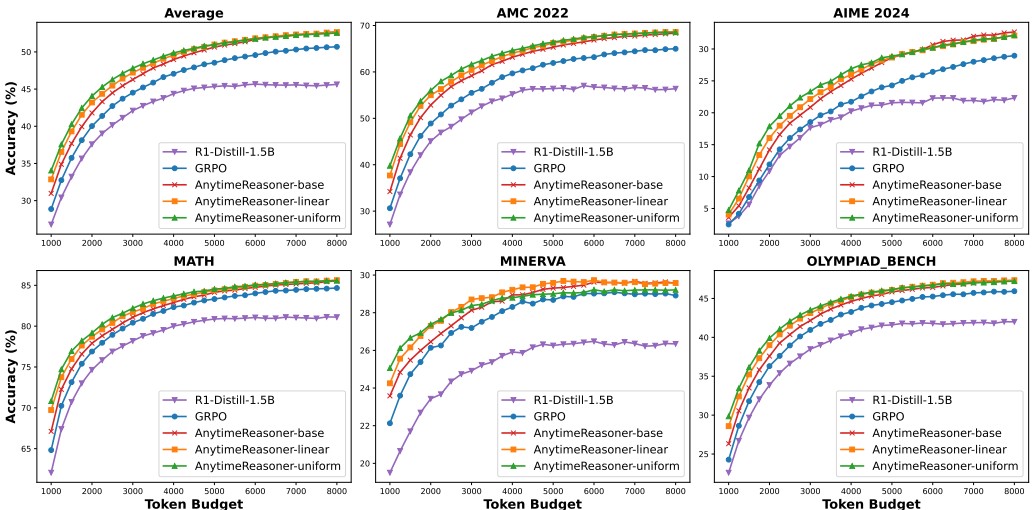

Figure 4: The comparison of anytime reasoning performance between GRPO and our *AnytimeReasoner* with various prior budget distributions. Notably, the accuracies at the maximum token budget (8000) reflect the performance in the standard reasoning task.

## 3 Experiments

We implement our algorithms based on the Verl framework [Sheng et al., 2024], incorporating several key modifications as detailed in Appendix B. We employ Proximal Policy Optimization (PPO) [Schulman et al., 2017] to optimize both thinking and summary policies. For the thinking policy, we use BRPO to compute the advantage function, as detailed in Section 2.2. During training, we allocate four token budgets ($m = 4$) for thinking: {2000, 4000, 6000, 8000}. For each question, we sample a group of 8 complete thinking processes (stopped either by *</think>* or when exceeding 8000 tokens). We sample 4 answers to calculate the average score at each thinking budget, which is used to compute the advantage function as in Dr. GRPO [Liu et al., 2025]. The summary length is restricted to 128 tokens. We extract the first answer and use a rule-based verifier to determine the 0/1 outcome reward. As detailed in Section 2.3, we employ different prior distributions for the thinking and summary policies. Unless otherwise specified, the prior distribution $p'_{\mathcal{B}}$ for the summary policy is set to a uniform distribution.

We fine-tuned DeepSeek-R1-Distill-Qwen-1.5B [Guo et al., 2025] on 40,315 math problems from DeepScaleR [Luo et al., 2025] for a single epoch, using a batch size of 64 questions per policy iteration. Our experiments were conducted on 8 NVIDIA A100 80G GPUs, with each experiment taking approximately 30 hours to complete (less than 10% overhead in total compared to GRPO). During training, we evaluate the average scores of AIME2024 and AMC2022 every 20 steps and report their performance curves, sampling 32 responses for each question. After training, we assess the final model using five benchmarks: AIME2024 [Li et al., 2024a], AMC2022 [Li et al., 2024a], MATH500 [Hendrycks et al., 2021], Minerva Math [Lewkowycz et al., 2022], and Olympiad Bench [He et al., 2024], with 32 uniform token budgets ranging from 0 to 8000. We compare our methods with GRPO [Shao et al., 2024], incorporating the corrections introduced in Dr. GRPO [Liu et al., 2025].

### 3.1 Main Results

We consider the following prior distributions $p_{\mathcal{B}}$ when optimizing the thinking policy by equation 3:

- *Base*: We only optimize the final performance as in standard reasoning task, namely $P_m = 1$.

- *Uniform*: We set $p_{\mathcal{B}}$ as a uniform distribution.

- *Linear*: We assign probability proportional to the budget length, such that $p_{\mathcal{B}}(b) \propto b$.

We evaluate the final models after training and plot the score curves under varying thinking budgets in Figure 4. For each question in AMC and AIME, we sample 320 thinking processes to compute the average score. For other datasets, we sample 80 thinking processes per question.

As shown in Figure 4, all variants of our method consistently outperform GRPO by a large margin across varying prior distributions. With small budgets, *AnytimeReasoner-uniform* excels by prioritizing optimization of these budgets. When the thinking budget is large, *AnytimeReasoner* with different prior distributions tends to converge to similar performance, demonstrating the robustness of our approach. Notably, even for *AnytimeReasoner-base*, where we optimize performance only under the maximum thinking budget as in the GRPO baseline, we still achieve significant better performance at all thinking budgets. This improvement is due to the decoupled optimization and our variance reduction technique (discussed further in Section 3.2.3). More details and additional evaluations on longer context can be found in Appendix D.

## 3.2 Ablations

To further investigate which aspects of our framework contribute to performance improvements, we conduct detailed ablations considering three factors: verifiable dense rewards (Section 3.2.1), decoupled optimization (Section 3.2.2), and variance reduction (Section 3.2.3). We report three metrics during training. *Anytime Accuracy*: the average accuracy over thinking budgets at {2000, 4000, 6000, 8000}. *Final Accuracy*: the accuracy at the maximum budget (8000). *Average Thinking Length*: the average thinking length under the maximum budget (8000).

### 3.2.1 Verifiable Dense Rewards

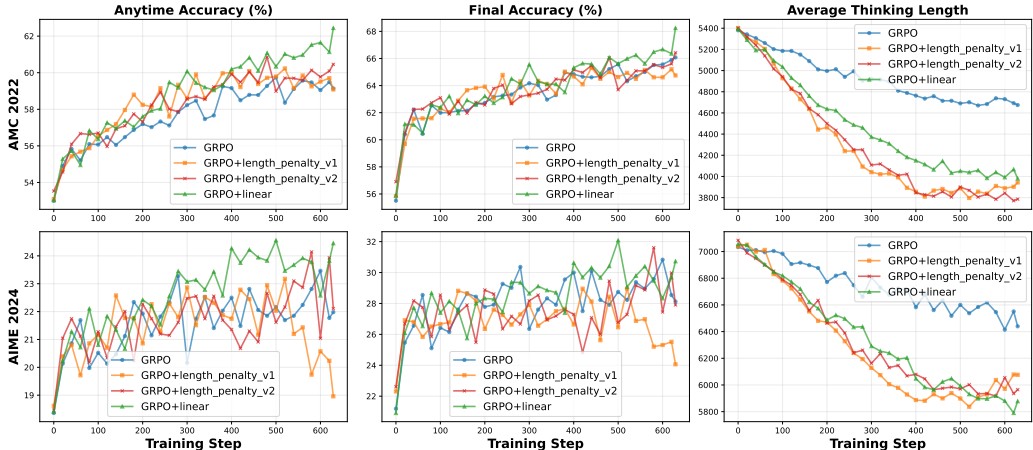

Figure 5: Ablation on verifiable dense rewards. For *GRPO+length_penalty_v1*, we follow Aggarwal and Welleck [2025], assigning reward $1 - \frac{0.2|z|}{b_m}$ for the correct answer and 0 for wrong answer. For *GRPO+length_penalty_v2*, we follow Arora and Zanette [2025] with $\alpha$ as 0.2.

We investigate the effectiveness of verifiable dense rewards by modifying the objective of the thinking policy in equation 3 with a *linear* prior distribution, while keeping the summary policy training consistent with GRPO. Specifically, we use $V_2$ as the variance reduction term to align with GRPO and eliminate the influence of enhanced variance reduction. To demonstrate our method's superior token efficiency, we compare it against reward shaping, which uses a length penalty on correct answers to encourage concise reasoning [Aggarwal and Welleck, 2025, Arora and Zanette, 2025].

As illustrated in Figure 5, incorporating dense rewards improves both the anytime and final performance. Notably, since our objective diverges from directly optimizing final performance as in the GRPO baseline, the observed improvements can be attributed to enhanced credit assignment facilitated by dense rewards. Another prominent observation is that the average thinking length is clearly shorter than the GRPO baseline under the maximum budget. This is because the thinking policy is encouraged to arrive at a correct answer as quickly as possible, making the model favor

shorter, correct responses. Although reward shaping with length penalty can also reduce the thinking length, it sacrifices the performance and is unstable during training.

### 3.2.2 Decoupled Optimization

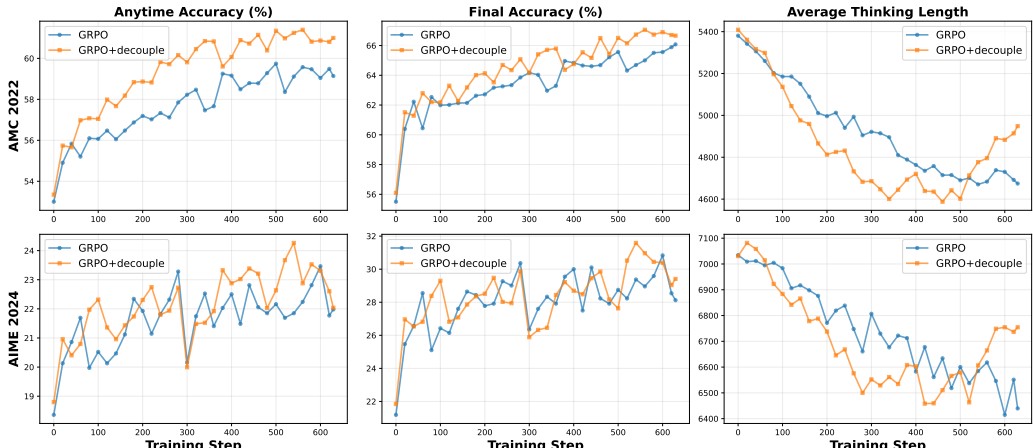

Figure 6: Ablation on decoupled optimization for summary policy.

To study the impact of decoupled optimization for thinking and summary policies (detailed in Section 2.3), we modify the training of summary policy in GRPO to align with *AnytimeReasoner*, while keeping the thinking policy training unchanged. Specifically, we sample 4 answers for each thinking budget in {2000, 4000, 6000, 8000}, applying GRPO within each summary group. This approach trains a summary policy under uniformly distributed thinking budgets, while the thinking policy optimizes performance only under the maximum budget (8000).

As shown in Figure 6, the decoupled GRPO clearly outperforms the vanilla GRPO, especially in the AMC benchmark. Notably, the significant improvement in anytime accuracy (the average score under sampled thinking budgets) indicates that decoupled optimization results in a better summary policy for anytime reasoning.

### 3.2.3 Variance Reduction

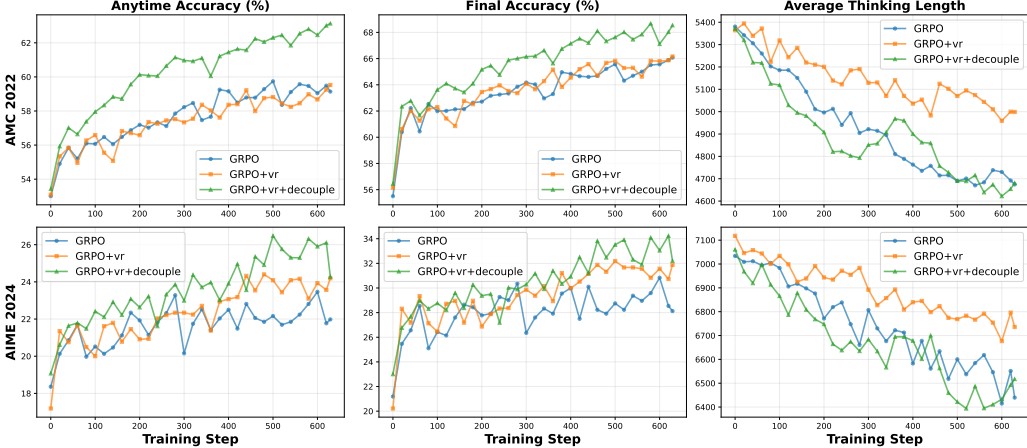

Figure 7: Ablation on variance reduction.

To evaluate the effectiveness of our BRPO variance reduction (as detailed in Section 2.2), we modified the training of the thinking policy by incorporating BRPO's variance reduction techniques, while maintaining the summary policy training consistent with GRPO. Specifically, we set $m = 4$ and $P(b_m) = 1$ in equation 7, aligning the objective exactly with GRPO.

Figure 7 shows that our approach enhances performance on the AIME benchmark. As discussed in Section 3.2.2, the suboptimal summary policy in GRPO may constrain the potential of BRPO's effectiveness. To address this, we introduced decoupled optimization (detailed in Section 2.3) to improve the summary policy, resulting in further performance gains.

### 3.3 Evaluation on 7B Model

We also evaluated our approach on a larger model, DeepSeek-R1-Distill-Qwen-7B. For this experiment, we modified the training setup by running two epochs on the DeepScaleR dataset with a batch size of 128 questions per iteration. We incorporated the *clip higher* technique from DAPO[Yu et al., 2025] to prevent entropy collapse[Cui et al., 2025b] observed in the training. As shown in Figure 8, our *AnytimeReasoner* framework achieves clearly superior performance in anytime reasoning, with a maximum improvement of about 5 absolute points. For standard reasoning, our methods outperform the GRPO baseline for most of the time during training, despite high variance in the final accuracy..

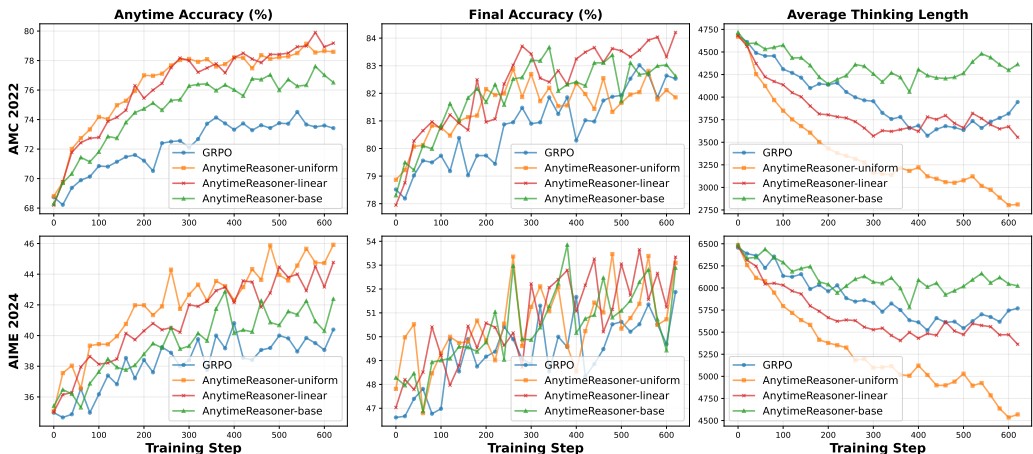

Figure 8: The training curves for DeepSeek-R1-Distill-Qwen-7B.

## 4   Related Works

**Reinforcement Learning with Verifiable Rewards**   Since the introduction of DeepSeek-R1 [Guo et al., 2025], a growing body of research has adopted the reinforcement learning with verifiable rewards (RLVR) paradigm [Lambert et al., 2024] to improve the reasoning capabilities of large language models (LLMs). SimpleRL [Zeng et al., 2025] provides the first open-source replication of R1-Zero in mathematical domains and analyzes RL dynamics across various base models. Hu et al. [2025] demonstrate that removing the KL regularization used in RLHF [Christiano et al., 2017] improves both RL efficiency and asymptotic performance. Liu et al. [2025] identify an optimization bias in GRPO [Shao et al., 2024] and propose Dr.,GRPO, which applies a Monte Carlo policy gradient method with a baseline [Sutton and Barto, 2018]. While these works improve our understanding of R1-Zero-style training, they still depend on sparse outcome-based rewards, which pose challenges for credit assignment and learning efficiency [Kazemnejad et al., 2024]. In contrast, our method introduces a novel policy optimization framework that leverages cheaply estimated *verifiable dense rewards* to improve sample efficiency and learning stability.

**Token Budget Efficiency of Reasoning Models**   Previous efforts have studied budgeted reasoning by reducing response length through prompting [Jin et al., 2024, Nayab et al., 2024, Lee et al., 2025, Ma et al., 2025] or adaptive sampling [Yang et al., 2025]. While these training-free approaches can shorten outputs, they often entail a trade-off between conciseness and task performance. More recent work explores token efficiency within online RL frameworks, enabling models to jointly optimize for accuracy and brevity. Yeo et al. [2025] observe that the output lengths on harder questions tend to grow during RL training, and propose a cosine-shaped reward to constrain length. Liu et al. [2025] trace this issue to optimization bias in GRPO and show that correcting it enhances token efficiency.

Further, Arora and Zanette [2025] and Aggarwal and Welleck [2025] apply explicit reward shaping to target shortened or fixed outputs. Our work differs by operating in an *anytime reasoning* framework, where the reasoning process can be interrupted at anytime and the best-effort solution should be provided [Dean and Boddy, 1988, Zilberstein and Russell, 1995]. Despite not explicitly enforcing conciseness, our objective naturally encourages efficient reasoning, as demonstrated empirically.

**Connection to MRT**   An independent work to ours, MRT [Qu et al., 2025], optimizes test-time compute by minimizing cumulative regret relative to an oracle. Since the oracle is unknown, they employ meta-RL [Xiang et al., 2025, Beck et al., 2023] as an approximation, aiming to maximize the "progress" of each newly generated *episode*. Despite sharing a similar high-level goal, our formulation fundamentally differs. Rather than minimizing regret, we optimize anytime performance by sampling the thinking budget from a prior distribution, remaining tractable with standard RL techniques. These foundational distinctions lead to significant methodological differences. Firstly, our approach operates on a per-token basis, instead of on *episode* which is ambiguous and can be hackable in RL if not well handled. Secondly, our method is grounded in principled RL, explicitly accounting for long-term returns. In contrast, MRT adopts a greedy strategy, optimizing the progress of immediate next episode only. Our experimental results also significantly outperform their reported outcomes. We achieve an accuracy of 32.7% compared to their reported 30.3% on AIME 2024.

## 5   Conclusion

The effectiveness of test-time scaling in LLM reasoning is commonly attributed to the generation-verification gap [Xiang et al., 2025], where verifying solutions is substantially easier than generating them. During reasoning, the model engages in an iterative search process, exploring potential solutions until a valid one is found. Once generated, the solution is verified for correctness, and this search-verification loop continues until a confident answer is produced.

In this work, we present a framework that systematically exploits this generation-verification gap. Our approach is based on the key observation that verifying answers and extracting them from partial reasoning traces is easy and computationally cheap. Building on this insight, we design our framework to produce answers at some predefined thinking budgets, thereby introducing verifiable dense rewards to enhance RL training. Furthermore, we utilize these additional rewards to construct a more effective variance reduction baseline than GRPO, significantly improving the stability and efficiency of RL training. By integrating these techniques, our framework achieves superior performance in both standard and anytime reasoning tasks.

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

# Appendix

## Table of Contents

## A    Implementation Details

We illustrate the implementation details about how we truncate the reasoning process and prompt the model to output an answer.

(a) Thinking is stopped by *</think>*.    (b) Thinking is stopped due to out of budget.

Figure 9: We decouple the generation of thinking and its summary. Given the question, the model first generates the thinking, which can be stopped by a special token *</think>* or the budget limit. Then we insert $**$ Final Answer $**$ (and two ellipsis $\cdots$ plus *</think>* for out of budget cases) to prompt the model to summarize the answer. In training, these inserted tokens will be ignored when calculating the loss.

## B    Tree-like Generation and Training

Unlike previous methods with sequential question-response generation and training, our approach employs a tree-like structure. In this section, we introduce how to address implementation challenges for efficient training.

During generation, we use the prefix caching feature of vLLM [Kwon et al., 2023] to reuse computations. We sample a complete thinking process $z$ for a question $x$, then split it based on predefined token budgets ($\{i, j, k\}$ in Figure 10). Each partial thinking process is appended with a special end-of-think token (*</think>*), and the model is prompted to output the answer directly (see Appendix A for more details).

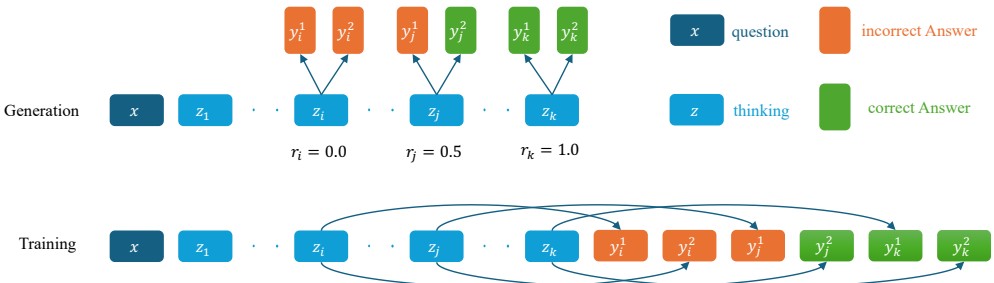

Figure 10: Our methods utilize a tree-like structure for generation and training.

During training, each response is typically concatenated with its corresponding question using FlashAttention [Dao et al., 2022] for speed. However, this introduces significant duplicated computation for tree-like structures, making it impractical due to high computational demands for LLM training. We implement a tree structure attention mask based on FlexAttention [Li et al., 2024b]. As shown in Figure 10, we append all summaries at the end of the thinking process and record their connection positions in a 1D tensor. This tensor is converted to a block mask by FlexAttention, avoiding 2D tensors that can cause out-of-memory issues for long generation lengths.

## C  Relation Between Standard and Anytime Reasoning

In this section, we provide a proof for the inequality below:

$$\mathcal{J}_{\text{anytime}}(\theta, \phi^*) \leq \mathcal{J}(\theta, \phi^*) \leq \frac{1}{P_m} \mathcal{J}_{\text{anytime}}(\theta, \phi^*).$$

According to equation 4, we have:

$$\mathbb{E}_{z \sim \pi_\theta(\cdot|x)} \left[ r_{\phi^*}(x, z_{\leq b}) \right] \leq \mathbb{E}_{z \sim \pi_\theta(\cdot|x)} \left[ r_{\phi^*}(x, z) \right],$$

Thus, it follows that:

$$
\begin{aligned}
\mathcal{J}_{\text{anytime}}(\theta, \phi^*) &= \mathbb{E}_{x \sim p_{\mathcal{X}}, z \sim \pi_\theta(\cdot|x)} \left[ E_{b \sim p_{\mathcal{B}}} \left[ r_\phi(x, z_{\leq b}) \right] \right] \\
&\leq \mathbb{E}_{x \sim p_{\mathcal{X}}, z \sim \pi_\theta(\cdot|x)} \left[ r_\phi(x, z) \right] \\
&= \mathcal{J}(\theta, \phi^*).
\end{aligned}
\tag{11}
$$

Assuming $r(x, y) \geq 0$, which is always achievable by adding a constant to each reward, we also have:

$$
\begin{aligned}
\mathcal{J}_{\text{anytime}}(\theta, \phi^*) &= \mathbb{E}_{x \sim p_{\mathcal{X}}, z \sim \pi_\theta(\cdot|x)} \left[ E_{b \sim p_{\mathcal{B}}} \left[ r_\phi(x, z_{\leq b}) \right] \right] \\
&\geq \mathbb{E}_{x \sim p_{\mathcal{X}}, z \sim \pi_\theta(\cdot|x)} \left[ P_m r_\phi(x, z_{\leq b_m}) \right] \\
&= \mathbb{E}_{x \sim p_{\mathcal{X}}, z \sim \pi_\theta(\cdot|x)} \left[ P_m r_\phi(x, z) \right] \\
&= P_m \mathcal{J}(\theta, \phi^*).
\end{aligned}
\tag{12}
$$

Combining 11 and 12, we can get

$$\mathcal{J}_{\text{anytime}}(\theta, \phi^*) \leq \mathcal{J}(\theta, \phi^*) \leq \frac{1}{P_m} \mathcal{J}_{\text{anytime}}(\theta, \phi^*).
\tag{13}$$

This completes the proof.

| Algorithm | AMC22 | AIME24 | MATH500 | Minerva | OlympiadBench | **Avg.** |
|---|---|---|---|---|---|---|
| R1-Distill-1.5B | 56.4 | 22.3 | 81.1 | 26.3 | 42.0 | 45.6 |
| GRPO | 65.0 | 28.9 | 84.7 | 28.9 | 45.9 | 50.7 |
| AR-base | 68.4 | **32.7** | 85.5 | **29.6** | **47.3** | **52.7** |
| AR-linear | **68.6** | 32.1 | **85.6** | **29.6** | **47.3** | 52.6 |
| AR-uniform | 68.5 | 32.2 | **85.6** | 29.2 | 47.2 | 52.5 |

Table 1: The **Final Accuracy** by evaluating the maximum budget (8000) for the final models.

| Algorithm | AMC22 | AIME24 | MATH500 | Minerva | OlympiadBench | **Avg.** |
|---|---|---|---|---|---|---|
| R1-Distill-1.5B | 48.2 | 16.3 | 74.5 | 24.1 | 36.0 | 39.8 |
| GRPO | 53.4 | 19.0 | 77.2 | 26.6 | 38.8 | 43.0 |
| AR-base | 57.0 | 21.9 | 78.2 | 27.3 | 40.2 | 44.9 |
| AR-linear | 58.2 | 22.3 | 79.0 | **27.7** | 40.9 | 45.6 |
| AR-uniform | **58.8** | **22.9** | **79.4** | 27.5 | **41.2** | **46.0** |

Table 2: The **Anytime Accuracy** by evaluating 32 budgets (every 250 tokens) for the final models.

# D   Experimental Results

## D.1   Main Results

We present the training curves of our *AnytimeReasoner* in Figure 11, corresponding to the experiments in Section 3.1. We also evaluate the performance of the models at training step of 600, and report the final accuracy in Table 1 and the anytime accuracy in Table 2.

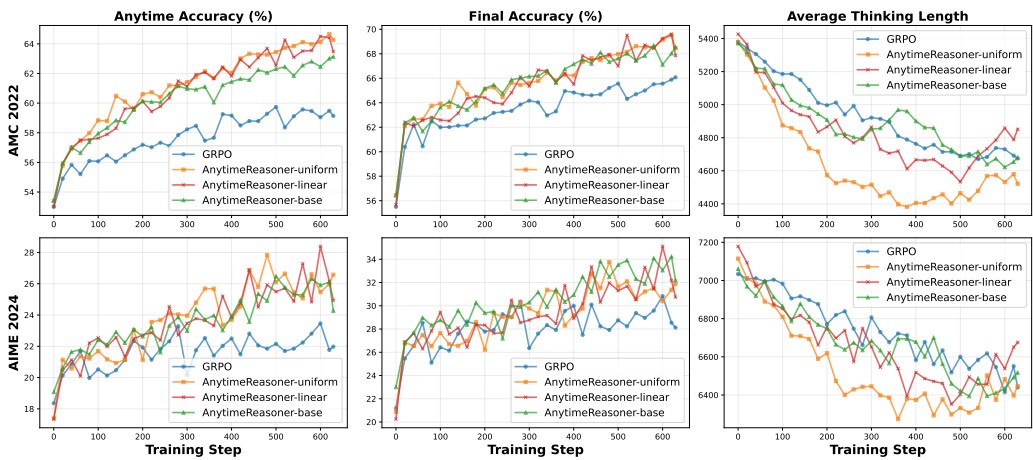

Figure 11: The training curves for main results.

## D.2   Evaluation on 16k Context Length

We also verified the effectiveness of our method on longer context lengths by fine-tuning DeepSeek-R1-Distill-Qwen-1.5B up to a 16k maximum context. We used a *uniform* prior over eight thinking budgets: {2k, 4k, 6k, 8k, 10k, 12k, 14k, 16k}. As shown in Figure 12, our approach consistently achieves stronger performance in both anytime and standard reasoning.

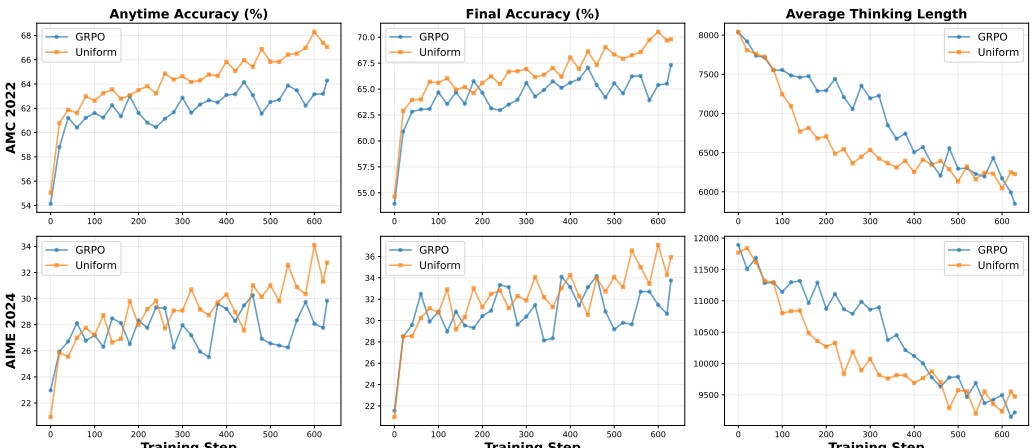

Figure 12: The training curves for DeepSeek-R1-Distill-Qwen-1.5B with maximum 16k context length.

