# OpenReview forum: "Optimizing Anytime Reasoning via Budget Relative Policy Optimization"
_NeurIPS.cc/2025/Conference — NeurIPS 2025 poster_

### Official Review · Reviewer_JDmN · 2025-06-23

**Clarity:** 3
**Significance:** 2
**Originality:** 3
**Rating:** 4
**Confidence:** 3

**Summary:**

The paper proposes an innovative reinforcement learning framework, AnytimeReasoner, which optimizes the real-time reasoning ability of LLM through budget sampling and dense reward mechanism. The core contributions include: budget sampling mechanism, BRPO variance reduction technology, and policy decoupling optimization. The paper verifies the superiority of the method on 6 mathematical reasoning datasets (AMC/AIME, etc.).

**Questions:**

1) Does the gradient derivation of formula 6 imply a KL constraint? If V(x,z_{<t}) is removed, will the strategy collapse?

2) Does the negative reward scenario (such as wrong answer penalty) destroy the inequality chain of formula 13? Additional theoretical analysis is needed

**Ethical Concerns:**

["NO or VERY MINOR ethics concerns only"]

**Quality:**

3

**Strengths And Weaknesses:**

Strengths:

1)  Dense Reward Mechanism: Generating verifiable intermediate rewards through mind truncation to solve the credit allocation problem caused by sparse rewards.

2) ​Performance Boundary Proof​​: Deriving the performance relationship between real-time reasoning and standard reasoning to provide a theoretical guarantee for the framework.

3) Tree-like attention mask (Figure 9): Solve the computational redundancy of traditional sequence training.


Weaknesses:

1) Convergence proof missing: BRPO's upper bound on gradient variance is not quantified (Formula 6)

2) Reward function assumptions: r(x,y)≥0 is required (Formula 12), and negative reward scenarios are not discussed

3) Limitation of baseline comparison: lack of comparison with classic RL algorithms such as PPO/REINFORCE

4) Insufficient domain generalization: only validates mathematical reasoning (algebra/geometry), does not cover scientific reasoning/decision making

5) Theoretical supplement: Add BRPO convergence proof (upper bound of gradient variance)

---

> ### Author Rebuttal · Authors · 2025-07-31
>
> We thank the reviewer for the valuable feedback. See response to individual points below.
>
> > Convergence proof missing: BRPO's upper bound on gradient variance is not quantified (Formula 6)
>
> > Theoretical supplement: Add BRPO convergence proof (upper bound of gradient variance)
>
> We don't understand what do you mean by convergence proof? To the best of our knowledge, we don't see such proof in any RL paper. Could you please provide a reference paper to clarify what is and why we need a "convergence proof"?
>
>
> > Reward function assumptions: r(x,y)≥0 is required (Formula 12), and negative reward scenarios are not discussed
>
> > Does the negative reward scenario (such as wrong answer penalty) destroy the inequality chain of formula 13? Additional theoretical analysis is needed
>
> Assuming $r\geq0$ (or satisfying some other contraints) is a common technique in RL to do theorectical proof, because in theory reward can be scaled/shifted without affecting the optimal policy in the MDP. In practice $r \geq 0$ is always achievable by adding a large constant to every reward, without changing the optimal policy. Please refer to [1] for a similar assumption and proof (see Assumption 4.1 and the proof of Theorem 4.1).
>
> In the case of Formula 12 and 13, the purpose is just to show that the final performance of the optimal anytime policy is well bounded, not too far from the optimal policy in standard reasoning task. The range and scale of rewards don't matter, not affecting this conclusion.
>
> Moreover, the right-hand side of formula 13 (formula 12) is relatively trivial. That’s why we only include the left-hand side (formula 11) in the main paper—it demonstrates that maximizing $J_{anytime}$ also improves performance on standard reasoning tasks. If formulas 12 is causing confusion, we can simply remove them without impacting the main logic of the paper.
>
> [1] Hong, Joey, Anca Dragan, and Sergey Levine. "Q-sft: Q-learning for language models via supervised fine-tuning." arXiv preprint arXiv:2411.05193 (2024).
>
>
> > Limitation of baseline comparison: lack of comparison with classic RL algorithms such as PPO/REINFORCE
>
> The main purpose of this paper is to show anytime reasoning framework is superior to standard reasoning. We argue that the choice of RL algorithm should be orthogonal to our main experiments, so a fair comparison should be: GRPO (under standard reasoning) vs. GRPO (under anytime reasoning), or PPO (under standard reasoning) vs. PPO (under anytime reasoning).
>
> As PPO with a critic model is known to be noisy and requires extra resources, we just choose GRPO, which is more popular and widely adopted as a strong baseline in community. Due to the compute constraints and limited human efforts, we cannot afford two RL algorithms in this paper. We leave the exploration of other RL algorithms in the future work.
>
> Compared to GRPO, REINFORCE is a relatively weak baseline due to its high variance.
>
>
> > Insufficient domain generalization: only validates mathematical reasoning (algebra/geometry), does not cover scientific reasoning/decision making
>
> We follow many prior work, including DR.GRPO[2], DeepScaleR[3], MRT[4], to validate the results on five commonly used benchmark. Could you please specify which datasts for scientific reasoning/decision making? We will consider to evaluate on them once compute resources are available.
>
> [2] Liu, Zichen, et al. "Understanding r1-zero-like training: A critical perspective." arXiv preprint arXiv:2503.20783 (2025).
>
> [3] Michael Luo, et al. "DeepScaleR: Surpassing O1-Preview with a 1.5B Model by Scaling RL"
>
> [4] Qu, Yuxiao, et al. "Optimizing test-time compute via meta reinforcement fine-tuning." arXiv preprint arXiv:2503.07572 (2025).
>
> > Does the gradient derivation of formula 6 imply a KL constraint? If V(x,z_{<t}) is removed, will the strategy collapse?
>
> No, formula 6 is just the common policy gradient in RL. KL constraint is not implied.
>
> If $V(x,z_{<t})$ is removed, formula 6 just fallbacks to REINFORCE. It is still a valid RL algorithm, but suffers from high gradient variance.
>
> In this paper, we do not assume whether KL constraint should be applied. It's a free choice up to the use cases. In our experiments, we remove the KL term as in many other prior work.

---

> > ### Author Response · Authors · 2025-08-06
> >
> > Dear Reviewer,
> >
> > We hope this message finds you well. As the discussion period is ending soon, we want to check if there are any remaining concerns or feedback you’d like us to address. If so, please let us know — your insights are invaluable, and we’re eager to further improve our work.
> >
> > Thank you again for your time and effort in reviewing our paper.

---

### Official Review · Reviewer_takS · 2025-07-01

**Clarity:** 4
**Significance:** 3
**Originality:** 3
**Rating:** 4
**Confidence:** 4

**Summary:**

This paper introduces AnytimeReasoner, a novel framework designed to optimize the "anytime reasoning" capabilities of LLMs under varying computational budgets. The core technique involves sampling thinking budgets from a prior distribution, truncating the reasoning trace at these budgets, and compelling the model to generate a summary. Overall this work proposes three key contributions:

- **Verifiable Dense Rewards in Anytime Reasoning**: This work introduces verifiable dense rewards throughout the anytime reasoning trajectory, which provides richer training signals and enables more effective credit assignment compared to the sparse rewards of conventional methods.

- **Budget Relative Policy Optimization (BRPO)**: A novel variance reduction technique that leverages these dense rewards to stabilize and improve the learning process.

- **Decoupled Optimization**: A strategy for separately optimizing the thinking and summary policies to ensure high-quality output from incomplete thoughts, resulting in a better summary policy for anytime reasoning.

**Questions:**

1. What is the computational overhead of the AnytimeReasoner framework during training compared to a vanilla GRPO implementation, especially considering the tree-like generation and multiple summary evaluations?
2. Have you considered evaluating the AnytimeReasoner framework on other foundational models like Llama, across different model scales, or extending to longer response length (>8k) ? These experiments may further validate the effectiveness of your method.
3. What is the intuition or theoretical explanation for using a linear prior distribution for budget sampling?

**Ethical Concerns:**

["NO or VERY MINOR ethics concerns only"]

**Limitations:**

See weakness.

**Paper Formatting Concerns:**

no concerns.

**Quality:**

3

**Strengths And Weaknesses:**

### Strength

1. This paper introduces a novel and practical concept of "anytime reasoning" to LLMs, addressing a practical problem where computational resources are limited or users are cost-sensitive, especially in online services.

2. This paper proposes a novel mechanism for introducing verifiable dense reward. By truncating the reasoning process and generating summaries at various budgets, the framework artfully transforms the sparse reward problem common in RL into a dense reward one. This provides a powerful signal for credit assignment and a richer way to analyze the model's reasoning process.
3. Through solid ablation studies, the paper validates the effectiveness of its three core components: verifiable dense rewards, decoupled optimization, and BRPO variance reduction. This clearly illustrates each component's contribution to the performance gains.


### Weakness

1. The study's findings are based on a single model architecture (DeepSeek-R1-Distill-Qwen-1.5B) and are validated exclusively on mathematical reasoning tasks. This narrow scope makes it unclear if the AnytimeReasoner framework would be equally effective for other foundational model series, reasoning domains, or for tasks requiring response lengths beyond the 8,000-token budget used in the experiments.

2. The framework introduces several new hyperparameters, such as the set of budget points $\{b_1,\cdots,b_B\}$ used in training , the discount factor $\lambda$ in BRPO , and the choice of prior distribution $p_B$. Although the paper tests a few distributions, it lacks an in-depth discussion of how these hyperparameters were chosen and how sensitive the model's performance is to them.

---

> ### Author Rebuttal · Authors · 2025-07-31
>
> We thank the reviewer for the insightful feedbacks. See responses to individual points below
>
> > The study's findings are based on a single model architecture (DeepSeek-R1-Distill-Qwen-1.5B) and are validated exclusively on mathematical reasoning tasks. This narrow scope makes it unclear if the AnytimeReasoner framework would be equally effective for other foundational model series, reasoning domains, or for tasks requiring response lengths beyond the 8,000-token budget used in the experiments.
>
> > Have you considered evaluating the AnytimeReasoner framework on other foundational models like Llama, across different model scales, or extending to longer response length (>8k) ? These experiments may further validate the effectiveness of your method.
>
> We believe RL algorithm should be orthogonal to model architecture because they are in totally different dimensions, so we mainly focus on the ablation study to verify the effectiveness of each component. We agree that it is worth to validate on more domains especially for coding. However, it is not realistic for most academic researchers to cover all of these scopes due to compute constraints and non-trivial engineering efforts. So we leave the exporation of other domains in future work.
>
> To address the concerns, we started several new experiments. (We report the best score during training in the table).
>
> ## R1-Distill-Qwen-1.5B with 16k token budget (completed)
>
> We achieve similar conclusion for 16k token budget: AR-uniform clearly outperforms GRPO in both anytime and final accuracy.
>
> | Algorithm    | Evaluation Dataset | Anytime Accuracy | Final Accuracy | Average Thinking Length |
> |--------------|-----------|------------------|----------------|------------------------|
> | GRPO         | AMC22     | 64.2             | 67.3           | ~6200                  |
> | AR-uniform   | AMC22     | 68.3             | 70.5           | ~6100                  |
> | GRPO         | AIME24    | 29.8             | 33.8           | ~9400                  |
> | AR-uniform   | AIME24    | 34.1             | 37.1           | ~9400                  |
>
> ## Qwen3-4B-Base with 16k token budget (Running)
>
> 400 training steps by now.
>
> After 160 step, GRPO fallbacks to short-CoT with an obvious performance drop, while AR-uniform still maintains a stable performance improvement. Even before 160 step, AR-uniform still clearly outperforms GRPO in both anytime and final accuracy. This shows that our framework leads to more stable RL training than GRPO.
>
> | Algorithm    | Evaluation Dataset | Anytime Accuracy | Final Accuracy |
> |--------------|-----------|------------------|----------------|
> | GRPO         | AMC22     | 52.5             | 55.6           |
> | AR-uniform   | AMC22     | 57.3             | 59.1           |
> | GRPO         | AIME24    | 21.6             | 22.7           |
> | AR-uniform   | AIME24    | 25.6             | 27.3           |
>
> ## R1-Distill-Qwen-7B with 16k token budget (Running)
>
> 300 training steps by now.
>
> | Algorithm    | Evaluation Dataset | Anytime Accuracy | Final Accuracy |
> |--------------|-----------|------------------|----------------|
> | GRPO         | AMC22     | 78.6             | 84.1           |
> | AR-uniform   | AMC22     | 80.6             | 84.4           |
> | GRPO         | AIME24    | 49.1             | 57.5           |
> | AR-uniform   | AIME24    | 51.4             | 59.2           |
>
> > The framework introduces several new hyperparameters... Although the paper tests a few distributions, it lacks an in-depth discussion of how these hyperparameters were chosen and how sensitive the model's performance is to them.
>
> Due to high computational requirements and the compute constrains, we do not tune the hyperparameters too much, the performance may not be optimal but we suppose in general it works well.
>
> For $b_B$, we always set it as the maximum allowed generation length, which is a commonly used hyperparameter in existing framework.
>
> For the support size $B$, we tried 2/4 for $b_B=8k$, found that both outperforms GRPO and $B=4$ performs better. We also tried $B=8$ for $b_B=16k$ (see above table), which also consistently outperforms GRPO. Generally, we set $b_i = 2000i$ in most of our experiments.
>
> For $\lambda$, we always set it as 0.5 in all of our experiments and didn't try other values. It may not be the optimal value regarding to variance reduction, but we suppose it should be better than GRPO generally because GRPO has high gradient variance for longer responses (see Figure 3). Additionally, it's easy and relatively cheap to check the variance reduction effectiveness before training (to decide the value of $\lambda$), as in Figure 3.
>
>
> > What is the computational overhead of the AnytimeReasoner framework during training compared to a vanilla GRPO implementation, especially considering the tree-like generation and multiple summary evaluations?
>
> Less than 10%.
>
> As mentioned in Appendix B, we implement a tree-like attention based on FlexAttention to avoid duplicated computation, which has comparable throughput with FlashAttention. So the only overhead is the extra summaries sampled. Ususally the summary length is much shorter than thinking, so the overhead is quite small (less than 10%). In our experiments, we sample 4 summaries for each budget (per 2k tokens), where the average length of summaries quickly converge to about 30. The overall training time for some AR settings are even faster than GRPO, because of the shorter response length.
>
> BTW, we have already uploaded our code in Supplementary Material (related code can be found in verl/models/transformers/flex_attn.py), and we will opensource all of the implementations once the paper is released. We hope our engineering efforts can also benefit the community for such tree-like attention training.
>
>
> > What is the intuition or theoretical explanation for using a linear prior distribution for budget sampling?
>
> The prior distribution describes the optimization objective for an anytime reasoning system, where the system can choose to interrupt the thinking process based on the workload, and users can also predefine a thinking budget to save cost. We can summarize the statistics into the prior distribution to optimize the overall performance of this anytime reasoning system.
>
> In the paper, we present 3 typical prior distributions:
> * the base prior which equals to the standard reasoning
> * the uniform prior which maximizes the area as in Figure 1 (easy to understand)
> * the linear prior which may be more realistic because longer responses are more likely to be interrupted
>
> Another motivation why we design the linear prior is: the uniform prior is not theoretically guaranteed to outperform the base prior in Final Accuracy (due to Eq. 5). So we design the linear prior as an interpolation between the uniform prior and base prior, aiming to maintain the Final Accuracy. However, experimental results surprisingly show that all 3 priors converges to similar Final Accuracy (see the last points in Figure 4), because BRPO and decoupled optimization already fully utilize the intermediate rewards to improve the performance.

---

> ### Author Response · Authors · 2025-08-04
> **Results of Further Evaluation**
>
> ## R1-Distill-Qwen-1.5B with 16k token budget (completed)
>
> | Algorithm    | Evaluation Dataset | Anytime Accuracy | Final Accuracy | Average Thinking Length |
> |--------------|-----------|------------------|----------------|------------------------|
> | GRPO         | AMC22     | 64.2             | 67.3           | ~6200                  |
> | AR-uniform   | AMC22     | 68.3             | 70.5           | ~6100                  |
> | GRPO         | AIME24    | 29.8             | 33.8           | ~9400                  |
> | AR-uniform   | AIME24    | 34.1             | 37.1           | ~9400                  |
>
> ## R1-Distill-Qwen-7B with 16k token budget (completed)
>
> | Algorithm    | Evaluation Dataset | Anytime Accuracy | Final Accuracy |
> |--------------|-----------|------------------|----------------|
> | GRPO         | AMC22     | 78.9             | 84.1           |
> | AR-uniform   | AMC22     | 83.4             | 86.0           |
> | GRPO         | AIME24    | 50.3             | 58.3           |
> | AR-uniform   | AIME24    | 54.6             | 61.0           |
>
> ## Qwen3-4B-Base with 16k token budget (completed)
>
> * After 160 step, GRPO fallbacks to short-CoT with an obvious performance drop, while AR-uniform still maintains a stable performance improvement.
> * Even before 160 step, AR-uniform still clearly outperforms GRPO in both anytime and final accuracy.
> * This shows that our framework leads to more stable RL training than GRPO.
>
> | Algorithm    | Evaluation Dataset | Anytime Accuracy | Final Accuracy |
> |--------------|-----------|------------------|----------------|
> | GRPO         | AMC22     | 52.5             | 55.6           |
> | AR-uniform   | AMC22     | 57.3             | 59.1           |
> | GRPO         | AIME24    | 21.6             | 22.7           |
> | AR-uniform   | AIME24    | 26.7             | 28.0           |

---

> > ### Author Response · Authors · 2025-08-06
> >
> > Dear Reviewer,
> >
> > We hope this message finds you well. As the discussion period is ending soon, we want to check if there are any remaining concerns or feedback you’d like us to address. If so, please let us know — your insights are invaluable, and we’re eager to further improve our work.
> >
> > Thank you again for your time and effort in reviewing our paper.

---

### Official Review · Reviewer_ATMH · 2025-07-02

**Clarity:** 3
**Significance:** 3
**Originality:** 3
**Rating:** 5
**Confidence:** 5

**Summary:**

This paper suggest anytime reasoning, a way to predefine the thinking budget and reason based on the budget. To implement this effectively, the author proposes to truncate the reasoning process based on the given budget and further propose budget relative policy optimization to reduce the variance.  The experimental result show that the propose method largely improve the efficiency and effectiveness over GPRO on anytime reasoning scenario.

**Questions:**

See the weakness above.

**Ethical Concerns:**

["NO or VERY MINOR ethics concerns only"]

**Final Justification:**

I still think the paper has the strength to be accepted. While there was no experimental support (so the concerns are still not resolved), I also agree that academia may not have enough resources for additional experiments. It is not necessary, but it would be great if the AC could check whether the authors' affiliations are academic.

**Limitations:**

See the weakness above.

**Paper Formatting Concerns:**

I have no concerns about paper formatting.

**Quality:**

3

**Strengths And Weaknesses:**

**Strength**

1) I believe this topic is important as overthinking (and thinking complexity) is one major problem of current reasoning models. In this sense, anytime reasoning is a novel topic to tackle this issue.

2) Also, the method is very well-designed. All components are reasonable and also somewhat simple.

3) The experiments are very well conducted. I like the fact that the authors have considered multiple benchmarks, including easy to hard datasets. Also, I like the analysis. All components are verified well.

4) Overall, the paper is well written and easy to follow

**Weakness**

I do not find a major weakness in this paper. I think the overall paper is very well written and should be accepted to NeurIPS.\
The following are some minor suggestions, and I hope these are well resolved throughout the rebuttal.

1) I can see that the proposed method is more effective on challenging datasets such as AMC or AIME. Does the author have any intuition behind this improvement? To validate this, I think it would be great if the author can provide the performance on easy dataset (e.g., GSM8K) and also include some challenging ones such as AMC 2023, AIME 2025 and GPQA diamond as well (I think it would have similar trends as AIME 2024 and AMC 2023, but will maybe worth it).

2) While I do agree that math is one reasoning-intensive domain, but also considering coding domains will benefit the readers, e.g., MBPP, Humaneval

3) Can the author provide the final accuracy based on no budget (i.e., generate until the model outputs the answer)? I think it is also good to know the upper bound performance without the constraints.

4) One concern is that the budget should be pre-defined before solving the problem. If there is a simple way to estimate the budget (i.e., complexity of the problem) before thinking, it would be great.

---

> ### Author Rebuttal · Authors · 2025-07-31
>
> We thank the reviewer for the insightful feedbacks. Please see individual points below.
>
> > I can see that the proposed method is more effective on challenging datasets such as AMC or AIME. Does the author have any intuition behind this improvement?
>
> Thanks for this insightful observation, we do observe similar trends. We guess there may be two reasons:
> * For harder dataset, the reward signal (the correct responses) is even sparser. By evaluating more thinking budgets, it can potentially provide more positive rewards to help RL training.
> * For harder problems, the model usually spends more thinking tokens before reaching a correct solution. As shown in Figure 3, GRPO suffers from high gradient variance when optimizing those long responses. By introducing BRPO, our RL training becomes more stable, which leads to better performance, especially for longer responses.
>
> > To validate this, I think it would be great if the author can provide the performance on easy dataset (e.g., GSM8K) and also include some challenging ones such as AMC 2023, AIME 2025 and GPQA diamond as well (I think it would have similar trends as AIME 2024 and AMC 2023, but will maybe worth it).
>
> > Can the author provide the final accuracy based on no budget (i.e., generate until the model outputs the answer)? I think it is also good to know the upper bound performance without the constraints.
>
> We thank the reviewer for the valuable suggestion. Due to the compute constraints, we didn't get enough resources to run such evaluation during rebuttal. We will run them when we get available compute resources.
>
>
> > While I do agree that math is one reasoning-intensive domain, but also considering coding domains will benefit the readers, e.g., MBPP, Humaneval
>
> We agree that it is worth to validate on more domains especially for coding. However, it is not realistic for most academic researchers to cover too much domains in one paper due to compute constraints and non-trivial engineering efforts. We leave the exporation of other domains in future work.
>
>
> > One concern is that the budget should be pre-defined before solving the problem. If there is a simple way to estimate the budget (i.e., complexity of the problem) before thinking, it would be great.
>
> To clarify, the generation of anytime reasoning is budget agnostic, taking only question $x$ as input (not depending on a predefined thinking budget $b$).
>
> For training, we believe it does help if we can adjust the pre-defined budget distribution based on the complexity of the problem. Some initial ideas: 1> we can use a reference model to estimate the token budget it requires to solve a problem 2> after we sample a group of responses, we can use the average response length (multiple a constant) to decide the maximum budget.
>
> In this paper, we want to keep it simple as the first version. We leave the exploration of adaptive budget in future work.

---

> > ### Comment · Reviewer_ATMH · 2025-08-01
> > **Thank you for the rebuttal**
> >
> > I thank the author for the rebuttal. The rebuttal is not fully convincing, but I do understand the resource constraint. I hope these suggestions are addressed in the final manuscript. I thank the reviewer again for the rebuttal. I will maintain my score.

---

> > > ### Author Response · Authors · 2025-08-07
> > >
> > > We thank the reviewer again for the insightful feedback. We got compute resources to evaluate some of the datasets you mentioned. We put the results below for your information.
> > >
> > > **I can see that the proposed method is more effective on challenging datasets such as AMC or AIME. Does the author have any intuition behind this improvement? To validate this, I think it would be great if the author can provide the performance on easy dataset (e.g., GSM8K) and also include some challenging ones such as AMC 2023, AIME 2025**
> > >
> > > As in the table below, we observe the most improvement in AMC2023, and the least improvement in GSM8k (with even slightly lower final accuracy). We obtain higher anytime accuracy (the average score under thinking budget {250, 500, ..., 8000}) in all these datasets, showing the effectiveness of our method to optimize the anytime reasoning.
> > >
> > > | Algorithm    | Dataset | Anytime Accuracy | Final Accuracy | Average Thinking Length |
> > > |--------------|-----------|------------------|----------------|-----------------------|
> > > | GRPO         | AMC23     | 60.2             | 73.2           | 4301                  |
> > > | AR-uniform   | AMC23     | 64.9 (+4.7)      | 76.1 (+2.9)    | 4255                  |
> > > | GRPO         | AIME25    | 17.6             | 23.7           | 6316                  |
> > > | AR-uniform   | AIME25    | 18.9 (+1.3)      | 24.3 (+0.6)    | 6315                  |
> > > | GRPO         | GSM8k     | 84.4             | 87.2           | 1416                  |
> > > | AR-uniform   | GSM8k     | 84.9 (+0.5)      | 86.8 (-0.4)    | 1230                  |
> > >
> > >
> > > **Can the author provide the final accuracy based on no budget (i.e., generate until the model outputs the answer)? I think it is also good to know the upper bound performance without the constraints.**
> > >
> > > Evaluating with a longer thinking budget than used during training relates to the topic of test-time compute **extrapolation**, as studied in [1]. While this is somewhat outside the main focus of our work, we are happy to provide this information. As shown in the tables below, our method demonstrates more stable extrapolation capability than GRPO, especially at the 24k thinking budget.
> > >
> > > Please note that the results is based on the model trained **under 8k maximum thinking budget**, which cannot represent the upper bound performance. When trained on 16k budget, we can obtain an obviously better performance than extrapolation, as reported under the comment of Reviewer takS. Unfortunately, we didn't save the checkpoints for 16k training. We will schedule another 16k training for further evaluation.
> > >
> > > [1] Setlur, Amrith, et al. "e3: Learning to Explore Enables Extrapolation of Test-Time Compute for LLMs." arXiv preprint arXiv:2506.09026 (2025).
> > >
> > > | Algorithm    | Dataset | Maximum Budget | Final Accuracy | Average Thinking Length | Anytime Accurarcy|
> > > |--------------|--------|-----------|---------|----------|---|
> > > | GRPO         | AMC23     | 8k  | 73.2           | 4301        | 60.2|
> > > | GRPO         | AMC23     | 16k | 75.4 (+2.2)    | 5228 (+927) | 67.7|
> > > | GRPO         | AMC23     | 24k | 75.0 (-0.4)    | 5387 (+159) | 70.8|
> > > | AR-uniform   | AMC23     | 8k  | 76.1           | 4255        | 64.9 (+4.7)|
> > > | AR-uniform   | AMC23     | 16k | 77.9 (+1.8)    | 5624 (+1369)| 72.2 (+4.5)|
> > > | AR-uniform   | AMC23     | 24k | 78.0 (+0.1)    | 5884 (+260) | 74.6 (+3.8)|
> > >
> > >
> > > | Algorithm    | Dataset | Maximum Budget | Final Accuracy | Average Thinking Length |Anytime Accurarcy|
> > > |--------------|-----------|----------|-------|---------|---|
> > > | GRPO         | AIME25    | 8k  | 23.7       | 6316        | 17.6|
> > > | GRPO         | AIME25    | 16k | 25.1 (+1.4)| 8506 (+2190)| 21.0|
> > > | GRPO         | AIME25    | 24k | 24.7 (-0.4)| 8941 (+435) | 22.3|
> > > | AR-uniform   | AIME25    | 8k  | 24.3       | 6315        | 18.9 (+1.3)|
> > > | AR-uniform   | AIME25    | 16k | 25.1 (+0.8)| 8696 (+2381)| 21.9 (+0.9)|
> > > | AR-uniform   | AIME25    | 24k | 25.6 (+0.5)| 9566 (+870) | 23.7 (+1.4)|
> > >
> > > **One concern is that the budget should be pre-defined before solving the problem. If there is a simple way to estimate the budget (i.e., complexity of the problem) before thinking, it would be great.**
> > >
> > > We would greatly appreciate it if you could let us know whether we have addressed your concern on this point.

---

### Official Review · Reviewer_Hybv · 2025-07-02

**Clarity:** 3
**Significance:** 2
**Originality:** 2
**Rating:** 3
**Confidence:** 4

**Summary:**

In this paper, the authors propose to improve the token efficiency of large language models by optimizing the accuracy of summarization across different token budgets with reinforcement learning. The authors introduce dense rewards uniformly spaced in the reasoning process. To ensure stable optimization with dense rewards using GRPO, the authors propose two variance reduction baselines, one considering rewards before some certain token budget, and one computing the average reward of all trajectories. The correlation analysis along with the variance analysis shows that the proposed techniques effectively reduces variances.

**Questions:**

1. In line 113 - 120, the claim "a larger thinking budget is supposed to yield better performance in expectation" is questionable. It is possible that an LLM could have higher accuracy under a restricted thinking budget than the full thinking budget. Therefore Eq. 4 and Eq. 5 may not hold in practice.

2. A natural counterpart for the introduced variance reduction techniques is using value models with PPO. How does GRPO with variance reduction compare with PPO w. value model?

3. In Figure 5, is "GRPO+Linear" using the variance reduction techniques? How does "GRPO+Uniform" that uses uniform distribution perform compared with "GRPO" and "GRPO+Linear" in this case?

4. In this paper, the authors use a summary policy to summarize a partial reasoning trajectory within 128 tokens. This design would allow the LLM learning to perform additional reasoning within the 128 tokens, contradicting the argument of "Budget-agnostic LLMs" in the MRT paper [1]. Could the authors provide a discussion on any negative effects of using and optimizing a summary policy?

[1] Qu, Yuxiao, et al. "Optimizing test-time compute via meta reinforcement fine-tuning." arXiv preprint arXiv:2503.07572 (2025).

**Ethical Concerns:**

["NO or VERY MINOR ethics concerns only"]

**Limitations:**

As I discussed in the Weakness, the major limitation of this work is lack of meaningful baselines, making it unable to make a fair comparison with prior works. It would be great if the authors could include more baselines.

**Paper Formatting Concerns:**

No major formatting issues are found in this paper.

**Quality:**

2

**Strengths And Weaknesses:**

Strength:
- The writing is clear and easy to follow.
- This work introduces a natural online extension to the prior work, Meta Reinforced Fine-Tuning, by introducing dense rewards.
- The proposed variance reduction techniques have shown to be able to reduce the variance by a clear margin.

Weakness:
- The major issue is lack of sufficient baselines. There are many prior works on improving the token efficiency through training, e.g. MRT [1], Efficient Reasoning [2], and O1-Pruner [3].
- There exists a convergence issue regarding the experiment setup. The GRPO baseline has not converged according the benchmark results and training curves. If the DeepSeek-R1-Distill-Qwen-1.5B model is well trained, it should achieve at least 40+ score on the AIME24 benchmark.
- Lack of evaluation on other LLMs. The experiments are all taken on the DeepSeek-R1-Distill-Qwen-1.5B model, limiting the scope of evaluation.

[1] Qu, Yuxiao, et al. "Optimizing test-time compute via meta reinforcement fine-tuning." arXiv preprint arXiv:2503.07572 (2025).
[2] Arora, Daman, and Andrea Zanette. "Training Language Models to Reason Efficiently." arXiv preprint arXiv:2502.04463 (2025).
[3] Luo, Haotian, et al. "O1-Pruner: Length-Harmonizing Fine-Tuning for O1-Like Reasoning Pruning" arXiv preprint arXiv:2501.12570v2 (2025).

---

> ### Author Rebuttal · Authors · 2025-07-31
>
> We thank the reviewer for the valuable comments. We respond to individual points from your review below.
>
> > In this paper, the authors propose to improve the token efficiency of large language models
>
> To clarify, while our method does improve the token efficiency in some sense, we believe that the motivation and purpose of the paper go far beyond just token efficiency.
>
> As stated in the paper, our primary goal to **improve the anytime reasoning capability**, conceptually similar to the established topic in computer science known as the **Anytime Algorithm** (see wikipedia for details). In our setting, reasoning can be interrupted at anytime (following a prior distribution $p_B$). When interrupted, the model should provide the best solution available and is expected to enhance the solution (in quality, accuracy, or confidence) when additional resources are allocated.
>
> In standard reasoning tasks, the model is not interruptible and always runs to completion under a fixed budget. In contrast, anytime reasoning provides more flexibility and supports features like predefined **thinking budget** (as seen in Gemini 2.5) for cost-sensitive users. Compared to budget-aware methods like L1 (which include the thinking budget in context before reasoning), anytime reasoning operates in a budget-agnostic manner. This supports an economical strategy by incrementally increasing the budget, allowing for continued thinking and reusing the computation already spent.
>
> > The major issue is lack of sufficient baselines. There are many prior works on improving the token efficiency through training, e.g. MRT [1], Efficient Reasoning [2], and O1-Pruner [3].
>
> For [2] and [3], they are basically based on reward shaping with length penalty. We do compare with a similar baseline L1-Max [4] in our Ablation 3.2.1 (experimental setting in line 221 and the results in line 228). Unfortunately, we uploaded the wrong Figure 5 by mistake, so the curve of length-penalty is missing. The experimental results of length-penalty are (we use the maximum value during training):
>
> | Algorithm    |  Dataset | Anytime Accuracy | Final Accuracy | Average Thinking Length |
> |----|------|------|-------|---------|
> | GRPO  | AMC22     | 59.5    | 66.0   | ~4700   |
> | GRPO+length_penalty   | AMC22     | 60.2    | 65.2  | ~3900 |
> | GRPO+linear  | AMC22     | 61.8    | 67.0  | ~4100  |
> | GRPO         | AIME24    | 23.5 | 30.5 | ~6600 |
> | GRPO+length_penalty   | AIME24    | 23.2    | 29.0  | ~5900 |
> | GRPO+linear  | AIME24     | 25.2    | 32.0  | ~5900    |
>
> Additionally, as discussed above, anytime reasoning is far beyond token efficiency, so our work is fundamentally different with those prior work focusing on reducing the response length (for token efficiency), including [2] and [3].
>
> MRT [1] is somewhat conceptually similar to our method. However, they hadn't released their code by the time of our submission (their GitHub was empty). They released some code on July 15, but it only included a brief guideline for their RL method. Upon reviewing the guideline, we found that it doesn't make any sense and is inconsistent with the algorithm described in their paper. Moreover, implementing MRT ourselves is not feasible due to missing details — the paper contains only a single paragraph on their RL approach. Therefore, we provide a detailed discussion and comparison with MRT in the final paragraph of Related Work.
>
> [4] Pranjal Aggarwal and Sean Welleck. L1: Controlling how long a reasoning model thinks with
> reinforcement learning. arXiv preprint arXiv: 2503.04697, 2025.
>
> > There exists a convergence issue regarding the experiment setup. The GRPO baseline has not converged according the benchmark results and training curves. If the DeepSeek-R1-Distill-Qwen-1.5B model is well trained, it should achieve at least 40+ score on the AIME24 benchmark.
>
> Many factors can affect the score, including grader, computational resources, maximum token budget, and so on so forth. We argue that only apple-to-apple comparison is meaningful.
>
> Our implementation basically follows DeepScaleR, with 8k thinking budget (for both training and evaluation) and a stricter grader for formatting.
> Some examples reporting AIME24 performance based on the same model:
> * DeepScaleR official: around 32 under 8k token budget.
> * DeepScaleR reproduce (in their issue #44): around 29 under 8k token budget.
> * MRT [1]: 28.7 for original model, and 29.8 for GRPO, under 16k token budget.
> * LASER [5]: 28.9 for original model, 31.5 for LASER_8192. For evaluation, 32k is used.
>
> All these numbers align with our results.
>
> [5] Liu, Wei, et al. "Learn to reason efficiently with adaptive length-based reward shaping." arXiv preprint arXiv:2505.15612 (2025).
>
> > Lack of evaluation on other LLMs. The experiments are all taken on the DeepSeek-R1-Distill-Qwen-1.5B model, limiting the scope of evaluation.
>
> We believe RL algorithm should be orthogonal to model architecture because they are in totally different dimensions. Due to compute constraints and non-trivial engineering efforts, we mainly focus on the ablation study to verify the effectiveness of each component in the paper.
>
> To address the concerns, we started two new experiments on other models. (We report the best score during training in the table).
>
> ## Qwen3-4B-Base with 16k token budget (Running)
>
> 400 training steps by now.
>
> After 160 step, GRPO fallbacks to short-CoT with an obvious performance drop, while AR-uniform still maintains a stable performance improvement. Even before 160 step, AR-uniform still clearly outperforms GRPO in both anytime and final accuracy. This shows that our framework leads to more stable RL training than GRPO.
>
> | Algorithm    |  Dataset | Anytime Accuracy | Final Accuracy |
> |-----|----|----|-----|
> | GRPO         | AMC22     | 52.5  | 55.6 |
> | AR-uniform   | AMC22     | 57.3  | 59.1 |
> | GRPO         | AIME24    | 21.6  | 22.7 |
> | AR-uniform   | AIME24    | 25.6  | 27.3 |
>
> ## R1-Distill-Qwen-7B with 16k token budget (Running)
>
> 300 training steps by now.
>
> | Algorithm    |  Dataset | Anytime Accuracy | Final Accuracy |
> |----|----|-----|-----|
> | GRPO         | AMC22     | 78.6 | 84.1 |
> | AR-uniform   | AMC22     | 80.6 | 84.4 |
> | GRPO         | AIME24    | 49.1 | 57.5 |
> | AR-uniform   | AIME24    | 51.4 | 59.2 |
>
>
> > In line 113 - 120, the claim "a larger thinking budget is supposed to yield better performance in expectation" is questionable. It is possible that an LLM could have higher accuracy under a restricted thinking budget than the full thinking budget. Therefore Eq. 4 and Eq. 5 may not hold in practice.
>
> * For optimal summary policy, there is no doubt that the claim is true. So we use a conservative $\phi^*$ in Eq. 4 and 5 to avoid controversy.
> * As shown in Figure 4, it also holds in practice. Especially after RL, the performance is smoothly increasing with larger thinking budget.
> * We belive that test-time scaling is based on this premise basically, that's the core idea in principle how we improve the reasoning through RL.
>
> > A natural counterpart for the introduced variance reduction techniques is using value models with PPO. How does GRPO with variance reduction compare with PPO w. value model?
>
> The main purpose of this paper is to show anytime reasoning framework is superior to standard reasoning. We argue that the choice of RL algorithm should be orthogonal to our main experiments, so a fair comparison should be: GRPO (under standard reasoning) vs. GRPO (under anytime reasoning), or PPO (under standard reasoning) vs. PPO (under anytime reasoning).
>
> As PPO with a critic model is known to be noisy and requires extra resources, we just choose GRPO, which is more popular and widely adopted as a strong baseline in community. Due to compute constraints and limited human efforts, we cannot afford two RL algorithms in this paper. We leave the exploration of other RL algorithms in the future work.
>
> > In Figure 5, is "GRPO+Linear" using the variance reduction techniques? How does "GRPO+Uniform" that uses uniform distribution perform compared with "GRPO" and "GRPO+Linear" in this case?
>
> As in line 218, we use V_2 only to eliminate the influence of enhanced variance reduction.
>
> We didn't get enough compute resources during rebuttal to run this experiment, but we believe it won't affect the conclusion.
>
> As we only train the final summary to align GRPO in this ablation, the summary policy is suboptimal. So we suppose GRPO+linear would be more stable (in Final Accuracy) as the objective is more closer to standard reasoning. This is why we choose linear instead of uniform prior.
>
> > In this paper, the authors use a summary policy to summarize a partial reasoning trajectory within 128 tokens. This design would allow the LLM learning to perform additional reasoning within the 128 tokens, contradicting the argument of "Budget-agnostic LLMs" in the MRT paper [1]. Could the authors provide a discussion on any negative effects of using and optimizing a summary policy?
>
> * As in Figure 8, we prompt the model to output the final answer directly after <\think>, and extract the first answer within \box{}, instead of the last answer as in many privious work.
> * If the model does continual reasoning (<\think> appears in $y$), we label it as incorrect answer.
> * We found that the typical format of $y$ is: the model will rephrase the question, and then output the answer. So we set the budget as 128 to allow this format.
> * The average summary length is about 30, which cannot do meaningful reasoning compared to 8k thinking budget.
> * As in our ablation (Section 3.2.2 and 3.3.3), optimizing summary policy is critical, we don't see any negative effects.
> * We don't understand why it contradicts "Budget-agnostic LLMs", could you please clarify on this?

---

> > ### Comment · Reviewer_Hybv · 2025-08-03
> >
> > First I would like to thank the author for pointing out the issue regarding the reproducibility issue of MRT. However, a large part of my concerns still remain unresolved.
> >
> > **1. Lack of Sufficient Baselines on Token Efficiency.**
> >
> > Though the author emphasizes token efficiency is not the main purpose of this work, it is clearly stated in the abstract:
> >
> > >  "AnytimeReasoner, to optimize anytime reasoning performance, which aims to **improve token efficiency** and the flexibility of reasoning under varying token budget constraints""
> >
> > It is, clearly, token efficiency is an important part of the purpose of the work and related baselines should be included.
> >
> > Although the authors include GRPO with length penalty as some sort of length rewards, prior works have shown that [1] and [2], as pointed out in my initial review, could work better than simple length penalty approaches. It is also wired that the authors try to weaken the role of token efficiency as one aspect of their contribution.
> >
> > **2. The convergence issue**
> >
> > The authors list a set of factors that may the converged performance and argue that they perform training within 8k token budget. However, the implementation does not follow prior works, making it difficult to compare this work against prior works. For instance, DeepScaleR and some works on 1.5B models [3] use at least 24K generation budget for training. It is also questionable why the authors do not follow prior works.
> >
> > **3. "Budget-agnostic LLM"**
> >
> > Budget-agnostic LLM is a concept introduced in MRT stating that the LLM should work well for multiple compute budgets to be able to  guarantee optimal for any test-time compute budget. In AnytimeReasoner, the summarization policy starts from a truncated response and generates some text along with the final answer. Since it still allows the summarization policy to generate 128 tokens, the summarization policy can perform reasoning within the additional tokens, violating the "Budget-agnostic" concept (Note that "reasoning" can indeed happen outside of the <think> </think> tags.) If it is true, according to the authors, that the summary only rephrases the question and produces the answer, it seems unnecessary to optimize the summary policy. If the summary policy benefits from training, this would indicate the summary policy is possibly budget-aware.
> >
> > **4. Question about the claim:**
> >
> > > a larger thinking budget is supposed to yield better performance in expectation
> >
> > Basically the author's response is "because the claim is true, the claim is true". This claim is clearly not hold in general because there exist counter-example that the LLM may finally find a wrong answer even when it finds the correct answer during the reasoning process. The authors are expected to fix this claim or provide theoretical justification (if any).
> >
> > In general, the evaluation of the work, including the evaluation scope and implmentation details, stilll remains largely unsatisfying.
> >
> > [1] Arora, Daman, and Andrea Zanette. "Training Language Models to Reason Efficiently." arXiv preprint arXiv:2502.04463 (2025).
> >
> > [2] Luo, Haotian, et al. "O1-Pruner: Length-Harmonizing Fine-Tuning for O1-Like Reasoning Pruning" arXiv preprint arXiv:2501.12570v2 (2025).
> >
> > [3] ProRL: Prolonged Reinforcement Learning Expands Reasoning Boundaries in Large Language Models

---

> ### Author Response · Authors · 2025-08-04
> **Clarification on the misunderstanding**
>
> We thank the reviewer for the response. We believe there exist some misunderstandings regarding several of the concerns raised, and we address these points below.
>
> **1. This claim is clearly not hold in general because there exist counter-example**
>
> The claim is "a larger thinking budget is supposed to yield better performance **in expectation**". While counter-examples do exist, they occur with low probability. For two random variables $x$ and $y$, $E[x] \leq E[y]$ doesn't suggest $x \leq y$, and vice versa.
>
> **2. Budget-agnostic vs. budget-aware**
>
> * The difference between `budget-agnostic` and `budget-aware` is, whether the reasoning process is depending on a predefined thinking budget. A `budget-agnostic` policy should be sampled from $$z \sim \pi_\theta(\cdot|x)$$, while a `budget-aware` policy is sampled from $$z \sim \pi_\theta(\cdot|x,b)$$
> * As in Eq. 3, our method follows the `budget-agnostic` setting.
> * If not training the summarization policy, the summary is unformatted (may not return the answer and do continual reasoning) and suboptimal (not able to exact the correct answer).
> * Even if it does violate the “budget-agnostic” concept, are there any actual side effects? If so, could you clarify what they are?
>
> **3. The convergence issue**
>
> * As listed in the rebuttal of Reviewer takS, **we did a complementary experiment under 16k thinking budget**, which also justify our method. We will include it in our updated manuscript.
> * DeepScaleR took **3,800 A100 hours**, and ProRL took **16,000 H100 hours**, (for one single training), which are unaffordable to most of academic researchers, including us.
> * **Both DeepScaleR and ProRL[3] are orthogonal to our work.** For example, DeepScaleR is basically GRPO + 3-stage curriculum learning. We can surely replace GRPO with our method, but such comparison is just endless with little insight, only making readers confusing.
> * **Many prior work use a relatively short response length** due to compute constraints. For example, L1[4] uses 4k context length. MRT uses 16k context length.
> * **The most important metrics in our work is Anytime Accuracy**, which is not reported by prior work. Re-evaluating the released model is also not reasonable, because summarizing truncated reasoning is not trained at all.
>
> [4] Pranjal Aggarwal and Sean Welleck. L1: Controlling how long a reasoning model thinks with
> reinforcement learning. arXiv preprint arXiv: 2503.04697, 2025.
>
> **4. Weaken the role of token efficiency as one aspect of their contribution**
>
> We are not weakening the role of token efficiency. As stated in the rebuttal, **our contribution goes far beyond token efficiency, and fundamentally differs with prior work including [1] and [2]**
>
> To clarify, we think the definition of token efficiency is not well unified in community. Improving token efficiency typically invovles reducing overthinking. Beyond reducing overthinking, it is a tradeoff between token (compute) and accuracy (performance), as suggested in Efficient Reasoning [1] and [5].
>
> In many prior work, including [1] and [2], they are trying to introduce length penalty to imporve the token efficiency by reducing the response length, while maintaining the accuracy or only with a small drop. Such methods, **cannot support anytime (or budgeted) reasoning**, so they can only plot one single point (for each training) in the Pareto efficiency curve of token-accuracy. In [1], they use different hyperparameters to control the penalty scale (thus multiple training) to show they can achieve a better token-accuracy tradeoff.
>
> Our anytime reasoning framework, **measures and improves token efficiency in a totally different manner**. As we support budgeted reasoning, we can efficiently evaluate the accuracy at every token budget (as shown in Figure 4). In an anytime reasoning system, anytime accuracy is a more important metrics than final accuracy and response length, because it measures the performance of overall system, and also "the token efficiency".
>
> [5] Gao, Jiaxuan, et al. "How Far Are We from Optimal Reasoning Efficiency?." arXiv preprint arXiv:2506.07104 (2025).
>
> **5. prior works have shown that [1] and [2], as pointed out in my initial review, could work better than simple length penalty approaches**
>
> * We do not see any clear evidence on this. Could you please provide some references?
> * As mentioned above, it is hard to decide the hyperparameter because it is always a tradeoff between token and accuracy.
>
> **6.In general, the evaluation of the work, including the evaluation scope and implmentation details, stilll remains largely unsatisfying.**
>
> We thank the reviewer for the feedback. In this work, we mainly focus on the **apple-to-apple ablation study** to provide more meaningful insights, **instead of reporting a SOTA score** (which requires more compute). We will extend the evaluation scope, including:
> * R1-Distill-Qwen-1.5B under 16k budget.
> * R1-Distill-Qwen-7B under 16k budget.
> * Qwen3-4B-Base under 16k budget.

---

> ### Author Response · Authors · 2025-08-04
> **Results of Further Evaluation**
>
> ## R1-Distill-Qwen-1.5B with 16k token budget (completed)
>
> | Algorithm    | Evaluation Dataset | Anytime Accuracy | Final Accuracy | Average Thinking Length |
> |--------------|-----------|------------------|----------------|------------------------|
> | GRPO         | AMC22     | 64.2             | 67.3           | ~6200                  |
> | AR-uniform   | AMC22     | 68.3             | 70.5           | ~6100                  |
> | GRPO         | AIME24    | 29.8             | 33.8           | ~9400                  |
> | AR-uniform   | AIME24    | 34.1             | 37.1           | ~9400                  |
>
> ## R1-Distill-Qwen-7B with 16k token budget (completed)
>
> | Algorithm    | Evaluation Dataset | Anytime Accuracy | Final Accuracy |
> |--------------|-----------|------------------|----------------|
> | GRPO         | AMC22     | 78.9             | 84.1           |
> | AR-uniform   | AMC22     | 83.4             | 86.0           |
> | GRPO         | AIME24    | 50.3             | 58.3           |
> | AR-uniform   | AIME24    | 54.6             | 61.0           |
>
> ## Qwen3-4B-Base with 16k token budget (completed)
>
> * After 160 step, GRPO fallbacks to short-CoT with an obvious performance drop, while AR-uniform still maintains a stable performance improvement.
> * Even before 160 step, AR-uniform still clearly outperforms GRPO in both anytime and final accuracy.
> * This shows that our framework leads to more stable RL training than GRPO.
>
> | Algorithm    | Evaluation Dataset | Anytime Accuracy | Final Accuracy |
> |--------------|-----------|------------------|----------------|
> | GRPO         | AMC22     | 52.5             | 55.6           |
> | AR-uniform   | AMC22     | 57.3             | 59.1           |
> | GRPO         | AIME24    | 21.6             | 22.7           |
> | AR-uniform   | AIME24    | 26.7             | 28.0           |

---

> > ### Author Response · Authors · 2025-08-06
> >
> > Dear Reviewer,
> >
> > We hope this message finds you well. As the discussion period is ending soon, we want to check if there are any remaining concerns or feedback you’d like us to address. If so, please let us know — your insights are invaluable, and we’re eager to further improve our work.
> >
> > Thank you again for your time and effort in reviewing our paper.

---

> > > ### Comment · Reviewer_Hybv · 2025-08-08
> > >
> > > Thank you for the response and new experiment results. However, my major concerns are still not resolved.
> > >
> > > First, regarding the token efficiency methods, though they do not directly enhance anytime accuracy, it is unknown how these methods would really perform unless a fair comparison is made. The submission lacks sufficient baselines for proper position against prior works.
> > >
> > > Second, the convergence issue makes it unclear whether the proposed method would be really better than GRPO on advanced reasoning models, especially given that the approach has higher computational cost than standard RLVR algorithms. Also, the new experiment results during the rebuttal period are still not taken with advanced models.
> > >
> > >
> > > With these considerations, I would maintain my current score.

---

> > > > ### Author Response · Authors · 2025-08-08
> > > >
> > > > We thank the reviewer for the response. While we respect your decision to maintain the score, we would like to offer the following clarifications.
> > > >
> > > > > **First, regarding the token efficiency methods, though they do not directly enhance anytime accuracy, it is unknown how these methods would really perform unless a fair comparison is made. The submission lacks sufficient baselines for proper position against prior works.**
> > > >
> > > > **We will include Efficient Reason [1] in the updated manuscript**. O1-Pruner [2] has an obvious overhead due to the additional rollouts of reference model, thus we just skip it.
> > > >
> > > > [1] Arora, Daman, and Andrea Zanette. "Training Language Models to Reason Efficiently." arXiv preprint arXiv:2502.04463 (2025).
> > > >
> > > > [2] Luo, Haotian, et al. "O1-Pruner: Length-Harmonizing Fine-Tuning for O1-Like Reasoning Pruning" arXiv preprint arXiv:2501.12570v2 (2025).
> > > >
> > > > > **the convergence issue makes it unclear whether the proposed method would be really better than GRPO on advanced reasoning models**
> > > >
> > > > As discussed above, **there is no convergence issue**. Using a shorter context length is a common setting in prior work. We also want to emphasize that **our contribution is algorithmic rather than releasing a SOTA model**. It is pointless to simply compare the scores.
> > > >
> > > > > **especially given that the approach has higher computational cost than standard RLVR algorithms**
> > > >
> > > > **The overall overhead of our approach is less than 10%.**
> > > >
> > > > As mentioned in Appendix B, we implement a tree-like attention based on FlexAttention to avoid duplicated computation, which has comparable throughput with FlashAttention. So the only overhead is the extra summaries sampled. Ususally the summary length is much shorter than thinking, so the overhead is quite small (less than 10%). In our experiments, we sample 4 summaries for each budget (per 2k tokens), where the average length of summaries quickly converge to about 30. The overall training time for some AR settings are even faster than GRPO, because of the shorter response length.
> > > >
> > > > We have already uploaded our code in Supplementary Material (related code can be found in verl/models/transformers/flex_attn.py), and we will opensource all of the implementations once the paper is released. We hope our engineering efforts can also benefit the community for such tree-like attention training.
> > > >
> > > > > **the new experiment results during the rebuttal period are still not taken with advanced models.**
> > > >
> > > > We’re unclear on what you mean by “advanced models.” From our perspective, **the Qwen3 series is already a recent and capable family of models**. Could you clarify which models you have in mind?
> > > >
> > > > -------------------------------------------
> > > >
> > > > Finally, we want to emphasize our contribution.
> > > > * We believe anytime reasoning is a novel and practical concept for LLM reasoning, especially when computational resources are limited or users are cost-sensitive in online services.
> > > > * We proposed a simple yet effective approach to introduce verifiable dense rewards into LLM reasoning, addressing the well-known credit assignment issue in RLVR. While MRT also targets a conceptual dense objective (minimizing regret), in practice it optimizes only a single greedy next-episode progress reward, which still remains sparse.
> > > > * We proposed BRPO, which, as acknowledged, reduces variance by a clear margin.
> > > > * We conducted a thorough ablation study to demonstrate the effectiveness of each component to help readers better understand our approach.
> > > >
> > > > Thank you again for your valuable time.

---

### Note · Authors · 2025-08-12

Dear AC and Reviewers,

Thank you again for taking the time to read and provide thoughtful feedback on our work. Below is a concise summary of our contributions and key rebuttal points.

>**Summary of contributions**

* We introduce anytime reasoning for LLMs — a practical and novel concept, especially when compute is limited or users are cost-sensitive in online services.
* We propose a simple yet effective method to introduce verifiable dense rewards into LLM reasoning, addressing the credit assignment issue in RLVR.
* We present BRPO, which leads to lower variance than GRPO.
* We conduct a thorough ablation study to demonstrate the effectiveness of each component and aid understanding of our approach.

>**Evaluation on other models**

Reviewers Hybv and takS requested evaluations on additional models and budgets beyond 8k. During rebuttal, we ran 3 complementary experiments (all with 16k budgets):

* R1-Distill-Qwen-1.5B
* R1-Distill-Qwen-7B
* Qwen3-4B-Base

Reviewer Hybv questioned that these are not "advanced models" (without specifying which models qualify). We respectfully disagree and welcome concrete suggestions.

>**Lack of sufficient baselines**

There appears to be some misalignment during the rebuttal and discussion period. Reviewer Hybv requested comparisons to two prior works (length-based reward shaping) for token efficiency. As **we already compare against a typical linear length penalty** (used in L1, DAPO, and Kimi-k1.5), we clarified this during rebuttal in case it was missed (because we uploaded a wrong Figure 5 by mistake).

During discussion, Reviewer Hybv stated that our baseline is weak (without evidences) and maintained this concern. While we respectfully disagree, we will include the additional baseline requested by the reviewer.

We also reiterate that our work is fundamentally different from length-based reward shaping, and our contributions go far beyond token efficiency. We do not believe this point should be grounds for rejection.

>**Weakness from Reviewer takS and JDmN**

* Reviewer takS: We believe we have addressed the weaknesses and questions.
* Reviewer JDmN: RL seems outside the reviewer’s core expertise (confidence 3). Several concerns are unreasonable and not central to our work — e.g., BRPO convergence proof, Formula 12 correctness, and whether Formula 6 implies a KL.

As we received no response from both reviewers during the discussion, we kindly ask AC to take these into consideration when making a decision.

---

### Decision · Program_Chairs · 2025-09-17

**Decision:**

Accept (poster)

**Comment:**

This paper presents an approach to optimize anytime performance of LLM reasoning. Reviewers generally like the paper, though I personally feel that the claims of "principled RL" are overblown. I believe that the direction of the paper is correct, but the comparisons are not quite extensive. We are accepting the paper conditioned making the paper on better comparisons in the final version of the paper.